# EMBO
## *reports*

# NPC1 enables cholesterol mobilization during long-term potentiation that can be restored in Niemann–Pick disease type C by CYP46A1 activation

Daniel N Mitroi[1] (iD), Guadalupe Pereyra-Gómez[1], Beatriz Soto-Huelin[1], Fernando Senovilla[1], Toshihide Kobayashi[2], Jose A Esteban[1] & María Dolores Ledesma[1,*] (iD)

## Abstract

NPC is a neurodegenerative disorder characterized by cholesterol accumulation in endolysosomal compartments. It is caused by mutations in the gene encoding NPC1, an endolysosomal protein mediating intracellular cholesterol trafficking. Cognitive and psychiatric alterations are hallmarks in NPC patients pointing to synaptic defects. However, the role of NPC1 in synapses has not been explored. We show that NPC1 is present in the postsynaptic compartment and is locally translated during LTP. A mutation in a region of the NPC1 gene commonly altered in NPC patients reduces NPC1 levels at synapses due to enhanced NPC1 protein degradation. This leads to shorter postsynaptic densities, increased synaptic cholesterol and impaired LTP in NPC1$^{nmf164}$ mice with cognitive deficits. NPC1 mediates cholesterol mobilization and enables surface delivery of CYP46A1 and GluA1 receptors necessary for LTP, which is defective in NPC1$^{nmf164}$ mice. Pharmacological activation of CYP46A1 normalizes synaptic levels of cholesterol, LTP and cognitive abilities, and extends life span of NPC1$^{nmf164}$ mice. Our results unveil NPC1 as a regulator of cholesterol dynamics in synapses contributing to synaptic plasticity, and provide a potential therapeutic strategy for NPC patients.

**Keywords** cholesterol; CYP46A1; Efavirenz; NPC1; synaptic plasticity
**Subject Categories** Membrane & Trafficking; Neuroscience

## Introduction

Cholesterol is a major component of neural cell membranes vital to normal brain function and is particularly abundant in synaptic membranes where cholesterol-rich microdomains influence a number of protein complexes [1,2]. Despite the relevance of cholesterol in synaptic functions, little is known about the molecular mechanisms by which it influences memory and learning processes, and how its levels and distribution are modulated at synapses.

Niemann–Pick disease type C (NPC) is a neurological disorder caused by mutations affecting the gene encoding NPC1 [3,4], a ubiquitous endolysosomal membrane protein involved in intracellular cholesterol transport [5,6]. A gradual loss of cognitive abilities and psychiatric alterations are frequent consequences of the disease [7], pointing to synaptic alterations as key pathological events. A biochemical hallmark of NPC is the accumulation of cholesterol in the endolysosomal compartment of cells [8]. Although endolysosomal lipid storage affects all cell types, neurons are the most vulnerable to NPC1 deficiency, suggesting a specialized function for this protein in neuronal cells. The observation that NPC1 is present not only in neuronal cell bodies, where late endosomes and lysosomes mainly reside, but also in distal axons and synaptosomes [9,10] further supports a distinct synaptic role for this protein not primarily involved in the degradative pathway.

Few studies have addressed the role of NPC1 in synapses, of which most have been performed in mice lacking NPC1, a model of the most aggressive early-onset forms of NPC [11]. The absence of NPC1 causes presynaptic defects [10,12] and changes in neurotransmitter levels [13]. However, the molecular mechanisms underlying these alterations have not been elucidated and the direct contribution of neuronal cholesterol changes is controversial [12,14]. Even less is known about the potential postsynaptic functions of NPC1, but impairment in certain forms of postsynaptic plasticity such as cerebellar long-term depression (LTD) and hippocampal LTP [15] has been observed in *Npc1*-null mice [16]. Lipid dynamics is a major determinant for the synaptic plasticity instrumental for learning and memory [17]. Activation of NMDA glutamate receptors during LTP, which is considered to be the molecular paradigm of memory, triggers cholesterol redistribution and reduction that drive postsynaptic delivery of AMPA receptors in hippocampal neurons [18].

1 Department of Molecular Neuropathology, Centro de Biología Molecular "Severo Ochoa" (CSIC-UAM), Madrid, Spain
2 Laboratoire de Biophotonique et Pharmacologie, Faculté de Pharmacie, Université de Strasbourg, Illkirch, France
 *Corresponding author. Tel: +34 9119 64535; E-mail: dledesma@cbm.csic.es

Consistently, synaptosomes undergo reversible cholesterol reduction upon NMDA-mediated excitatory neurotransmission [19]. While the mechanism underlying synaptic activity-induced redistribution of cholesterol is not known, that of cholesterol reduction seems to be mediated by the cholesterol hydroxylase cytochrome P450 46A1 (CYP46A1). CYP46A1 is the enzyme responsible for cholesterol turnover in neurons [20] and mobilizes towards the plasma membrane upon synaptic stimulus by unclear means [21].

In the current work, we have used a recently generated mouse line ($NPC1^{nmf164}$), carrying a D1005G mutation in $Npc1$, similar to mutations commonly witnessed in human cases. $NPC1^{nmf164}$ mice develop comparable symptoms but with slower progression than $Npc1$-null mice, and thus, they are considered a more suitable model for late-onset NPC [22]. These mice have allowed us to evidence the role of NPC1 in LTP-induced cholesterol changes and CYP46A1 and AMPA receptor surface delivery, along with the altered synaptic composition/function and impaired memory when NPC1 is deficient. These anomalies could be prevented *in vitro* and *in vivo* by pharmacological activation of CYP46A1.

## Results

### NPC1 is present at the synapses where it is locally translated during LTP

Although the presence of NPC1 has been reported in synaptosomes [10], its distribution in synapses has not been addressed in detail. Thus, confocal microscopy and electron microscopy were used together with biochemical techniques to localize NPC1 in mouse primary neuronal cultures and brain tissue. Immunocytochemical analysis of mature hippocampal neurons from wild-type (wt) mice indicated the presence of NPC1 in both pre- and postsynaptic terminals, with a relative enrichment in the latter. Co-localization of NPC1 with presynaptic (synaptophysin) and postsynaptic (PSD95) markers was found to be 22.5 and 59%, respectively (Fig 1A). The postsynaptic localization of NPC1 was confirmed by electron microscopy in the hippocampus of wt mice using two different antibodies against the protein (Fig 1B and Appendix Fig S1). Although the

fixation protocols required for the immunoelectron microscopy hindered membrane visualization, NPC1 immunogold particles were found associated with membrane-like structures.

Western blot against NPC1 in total and synaptosomal extracts from wt mice also demonstrated the presence of this protein in synaptic membranes (Fig 1C and D). Antibody specificity was confirmed by the lack of signal in brain extracts from NPC1 knockout mice (Fig 1E). Quantitative PCR of the synaptosomal extracts from wt mice, in which nuclear contamination was ruled out (Appendix Fig S2), evidenced the presence of NPC1 mRNA in synapses (Fig 1F). To determine whether the presence of NPC1 mRNA in synapses resulted in the local synthesis of the protein during LTP, this form of plasticity was chemically induced in synaptosomes by adding glycine and KCl in the absence of $MgCl_2$ [23,24]. The efficiency of the chemical LTP (cLTP) protocol in synaptosomes was confirmed by the phosphorylation of the AMPA receptor subunit GluA1 [25] (Appendix Fig S3). Consistent with local NPC1 translation, cLTP in wt synaptosomes induced a significant 82% increase in NPC1 levels, which was prevented by incubation with the protein synthesis inhibitor cycloheximide (Fig 1G).

### The D1005G mutation in NPC1 decreases basal and LTP-induced levels of the protein in synapses due to enhanced degradation, and leads to altered synaptic morphology and high cholesterol levels

Western blot analysis of NPC1 expression in total brain extracts and synaptosomes from $NPC1^{nmf164}$ mice, which bear the D1005G mutation in the $Npc1$ gene, showed that while NPC1 levels were not significantly altered in total brain extracts (Fig 1C), the synaptic levels were drastically (70%) reduced compared to wt mice (Fig 1D and Appendix Fig S4). In contrast, the levels of other synaptic proteins such as synaptophysin, PSD95, Syntaxin 1A and VAMP2 were not altered (Appendix Fig S5).To determine whether the low synaptic NPC1 expression could be due to impaired transport of the protein when mutated, we expressed wt and D1005G NPC1 linked to the green fluorescent protein (GFP) in organotypic slice cultures of wt mice using Sindbis virus. Mobility associated with GFP-wtNPC1 and GFP-D1005GNPC1 particles was video-recorded in neurites

**Figure 1. NPC1 localization and expression levels in wt and $NPC1^{nmf164}$ mice.**

A   Single greyscale and coloured merged images of immunocytochemical staining against PSD-95, synaptophysin-1 (SYP1) and NPC1 proteins in a cultured hippocampal neuron from a wt mouse. Graphs show mean $\pm$ SEM of percentage of co-localization of NPC1 with PSD95 or SYP1 ($n$ = 12 neurons from three different cultures).

B   Electron microscopy image of immunogold labelling against NPC1 in the CA1 hippocampal region of a wt mouse. Presynaptic compartments are indicated in pink while postsynaptic in blue. White arrows show the 15-nm gold particles linked to NPC1 (d—dendrite, sv—synaptic vesicles, m—mitochondria).

C, D   Western blots against NPC1 and actin-β (ACTB) in total brain (C) and synaptosomes (D) from wt and $NPC1^{nmf164}$ mice. Graphs show mean $\pm$ SEM NPC1 level normalized to ACTB as a percentage of control (wt) values ($n$ = 4 mice, 3 months old, unpaired Student's *t*-test, **$P_{synaptosomes}$ = 0.0037).

E   Western blots against NPC1 and actin-β (ACTB) in total brain extracts from wt and *Npc1-null* mice (NPC1KO) containing the same amount of protein ($n$ = 2 mice, 6 weeks old).

F   Graph shows mean $\pm$ SEM mRNA levels of NPC1 in total extracts and in synaptosomes from wt and $NPC1^{nmf164}$ mice ($n$ = 4 mice, 3 months old).

G   Western blots against NPC1 and actin-β (ACTB) in synaptosomes from wt and $NPC1^{nmf164}$ mice in which cLTP was induced or not in the presence or absence of the protein synthesis inhibitor cycloheximide (CHX). Graphs show mean $\pm$ SEM NPC1 level normalized to ACTB as a percentage of control (wt non-cLTP-induced) values ($n$ = 3 mice, 3 months old, 2-way ANOVA, ***$P < 0.001$).

H   To the left Western blots against NPC1 and actin-β (ACTB) in synaptosomal extracts from wt and $NPC1^{nmf164}$ mice used as input for the immunoprecipitation assays shown in the right. Immunoprecipitates were pulled down with an anti-ubiquitin antibody and without it for negative control and analysed by Western blot against NPC1. MG132 was used as proteasome inhibitor. Graphs show mean $\pm$ SEM levels of NPC1 normalized to ACTB in the inputs (left) ($n$ = 3, one-way ANOVA, *$P_{WT+MG-132}$ = 0.0496, *$P_{NPC1nmf1+MG-132}$ = 0.0358) and of NPC1 associated with ubiquitin (right) ($n$ = 3 mice, 3 months old, one-way ANOVA, *$P_{NPC1nmf164}$ = 0.0237, *$P_{NPC1nmf164+MG-132}$ = 0.0386).

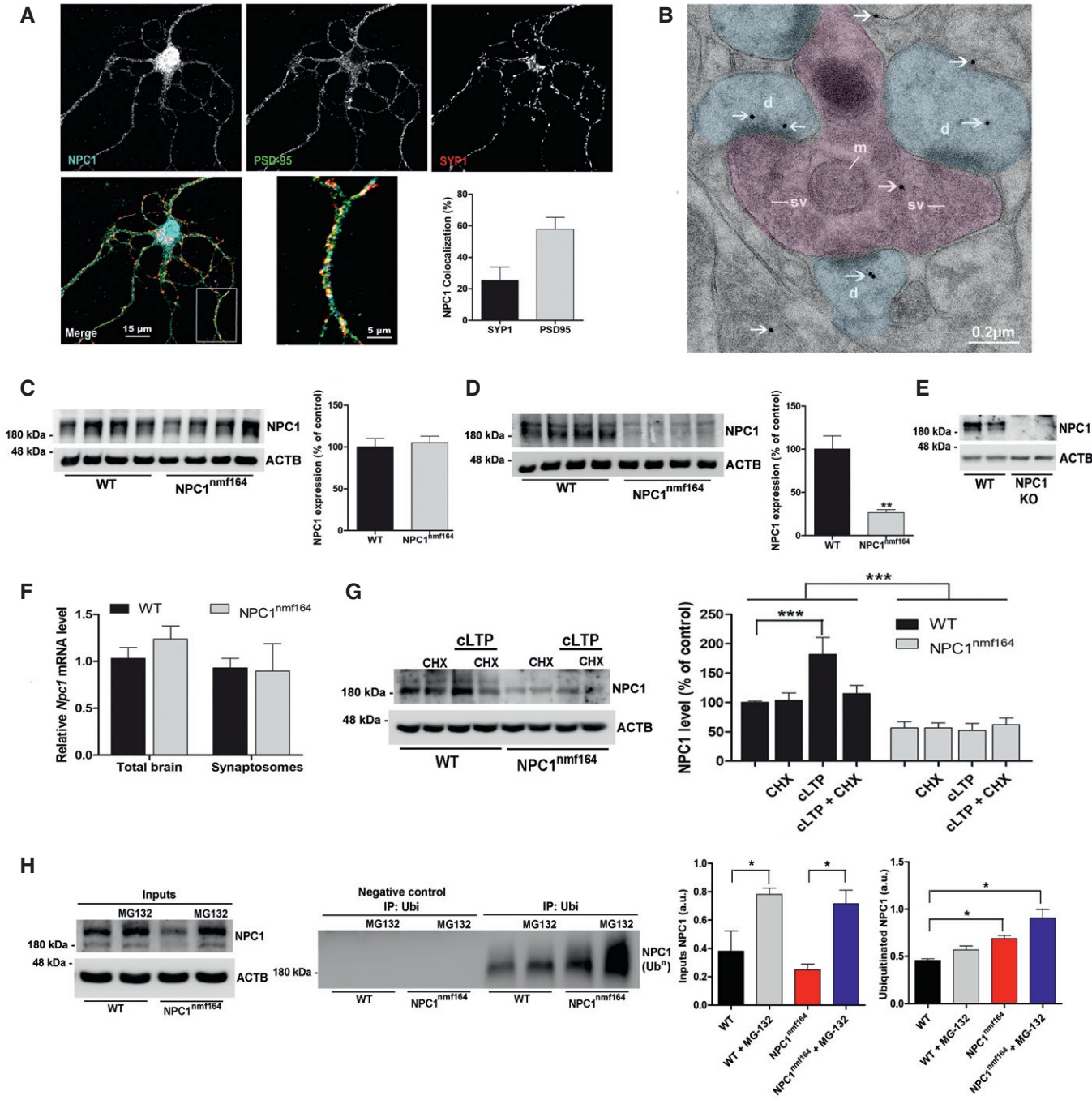

**Figure 1.**

(Movies EV1 and EV2), and their velocity was expressed as μm per second (Appendix Fig S6A). No significant differences were observed. The lower levels of expression achieved with GFP-D1005GNPC1 compared to GFP-wtNPC1 upon Sindbis virus infection (Appendix Fig S6B) argued in favour of increased degradation of the mutated NPC1. Since one of the major degradation pathways for NPC1 is via the proteasome [26–28], we measured the levels of ubiquitinated NPC1 in synaptosomes from wt and $NPC1^{nmf164}$ mice in the presence or absence of the proteasome inhibitor MG-132. Consistent with enhanced degradation of the mutant NPC1 by the proteasome,

we found its ubiquitination levels were higher and further increased upon proteasome inhibition, in $NPC1^{nmf164}$ compared to wt synaptosomes (Fig 1H). In agreement with enhanced degradation of D1005GNPC1, induction of cLTP failed to increase NPC1 levels in $NPC1^{nmf164}$ synaptosomes in the presence or absence of protein synthesis inhibitors in contrast to wtNPC1 (Fig 1G). NPC1 mRNA levels were not altered in $NPC1^{nmf164}$ synaptosomes (Fig 1F).

The reduced levels of NPC1 in the synapses of $NPC1^{nmf164}$ mice moved us to analyse synaptic features in the hippocampal CA1 region. Morphological analysis of asymmetric excitatory synapses

by electron microscopy (Fig 2A) revealed an average 25% reduction in the number of synapses per unit area in 3-month-old *NPC1*<sup>nmf164</sup> mice compared to age-matched wt mice (Fig 2B). The number of synaptic vesicles was also reduced by 51.4%, while their size was increased by 62% (Fig 2C and D). Alterations in the postsynaptic compartment were also found: Length of the postsynaptic density was 35.8% shorter, while its thickness was increased by 42.9% in the *NPC1*<sup>nmf164</sup> mice (Fig 2E and F). Quantification of cholesterol by enzymatic assays showed a mild but significant 16.5% increase in cholesterol levels in synaptosomes from *NPC1*<sup>nmf164</sup> mice compared to wt mice (Fig 2G).

### Altered synaptic function and behaviour in *NPC1*<sup>nmf164</sup> mice

The morphological alterations and the impaired LTP-induced NPC1 synthesis found in the *NPC1*<sup>nmf164</sup> synapses prompted us to analyse function. We relied on electrophysiological recordings from acute hippocampal slices to monitor excitatory synaptic responses from Schaffer collaterals to CA1 neurons. Basal synaptic transmission was increased in *NPC1*<sup>nmf164</sup> mice, while the fibre volley amplitude was not altered compared to wt mice (Fig 3A and B). Paired pulse facilitation (PPF) was increased (Fig 3C), suggesting impaired neurotransmitter release in *NPC1*<sup>nmf164</sup> mice. Importantly, and in

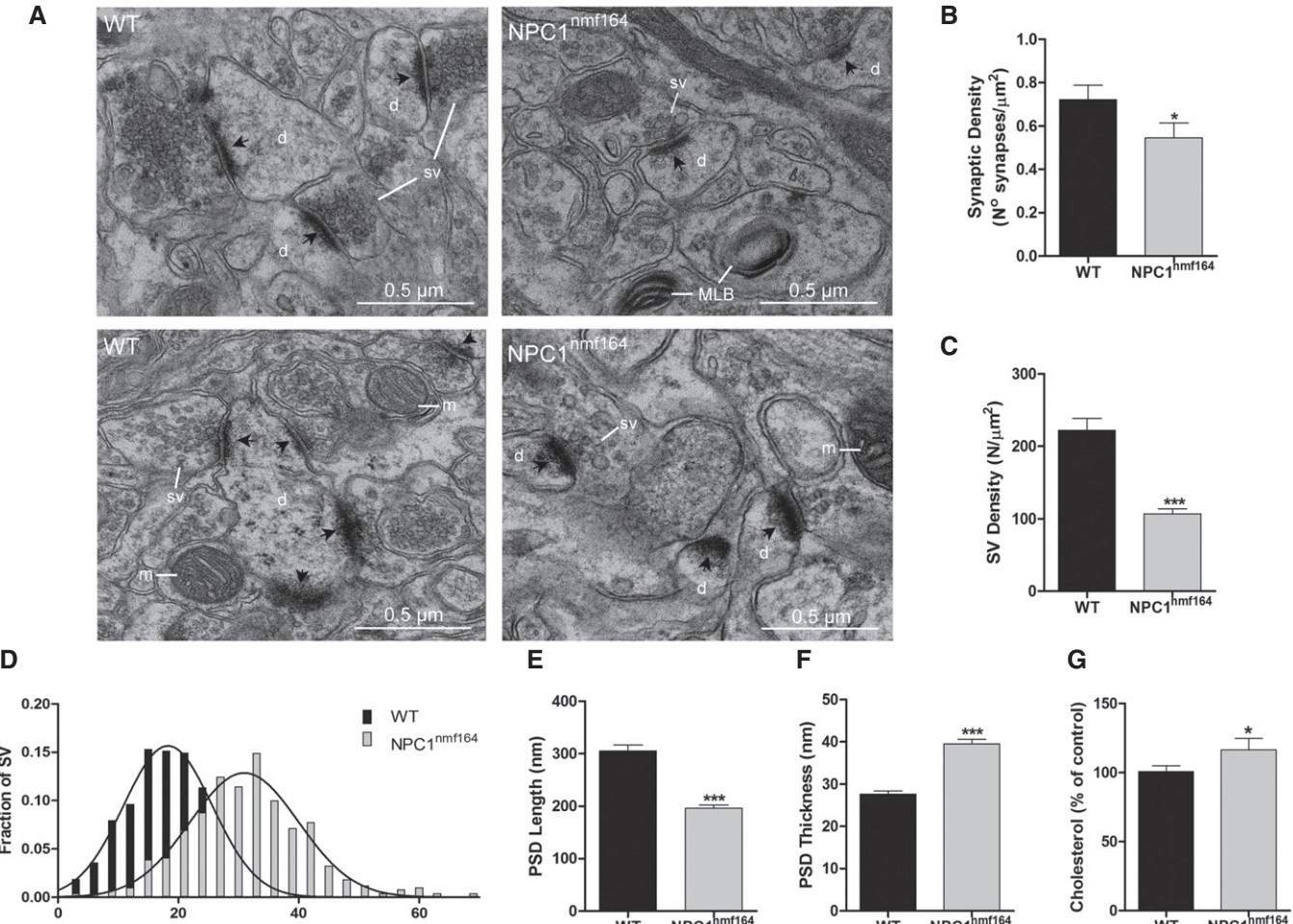

**Figure 2. Morphology and cholesterol level of synapses in *NPC1*<sup>nmf164</sup> mice.**

A  Representative electron micrographs of synapses in the hippocampal CA1 region in wt and *NPC1*<sup>nmf164</sup> mice (black arrows indicate postsynaptic densities, d—dendrite, sv—synaptic vesicles, m—mitochondria, MLB—multilamellar bodies).
B  Mean ± SEM synapse density (*n* = 3 mice, 3 months old, unpaired Student's *t*-test, *$P$ = 0.0377).
C  Mean ± SEM synaptic vesicle density (3,088 vesicles from 101 wt and 121 *NPC1*<sup>nmf164</sup> synapses from three different mice (3 months old) per group, unpaired Student's *t*-test, ***$P$ < 0.0001).
D  Synaptic vesicle diameter (528 wt and 507 *NPC1*<sup>nmf164</sup> vesicles from three different mice (3 months old) per group).
E  Mean ± SEM postsynaptic density length (101 wt and 121 *NPC1*<sup>nmf164</sup> postsynaptic densities from three different mice (3 months old) per group, unpaired Student's *t*-test, ***$P$ < 0.0001).
F  Mean ± SEM postsynaptic density thickness (*n* = 3 mice, 3 months old, unpaired Student's *t*-test, ***$P$ < 0.0001).
G  Mean ± SEM cholesterol levels in synaptosomes (*n* = 4 mice, 3 months old, unpaired Student's *t*-test, *$P$ = 0.0476) from wt and *NPC1*<sup>nmf164</sup> mice expressed as percentage of wt mice.

agreement with the impaired response regarding NPC1 synthesis during LTP (Fig 1F), this postsynaptic form of plasticity was virtually abolished in *NPC1^nmf164* mice (Fig 3D). In contrast, LTD was conserved (Fig 3E).

Given these alterations in synaptic plasticity forms that control memory and learning, several tests were conducted to monitor cognitive abilities in NPC1^nmf164 mice. In object placement recognition tests, the *NPC1^nmf164* mice poorly recognized the object in the novel location, indicative of impaired hippocampal spatial learning

and memory (Fig 3F). Consistently, in Y maze tests, *NPC1^nmf164* mice entered the novel arm less frequently (31.9%) than wt mice (43.1%) (Fig 3G). Associative learning and memory in the contextual fear conditioning test was also impaired in the *NPC1^nmf164* mice compared to wt mice, as evidenced by a 62% reduction in freezing time (Fig 3H). To rule out that impaired locomotor activity had affected the outcome of the aforementioned tests, exploratory activity was analysed by the open-field test, finding no significant difference in the distance covered by *NPC1^nmf164* and wt mice (Fig 3I).

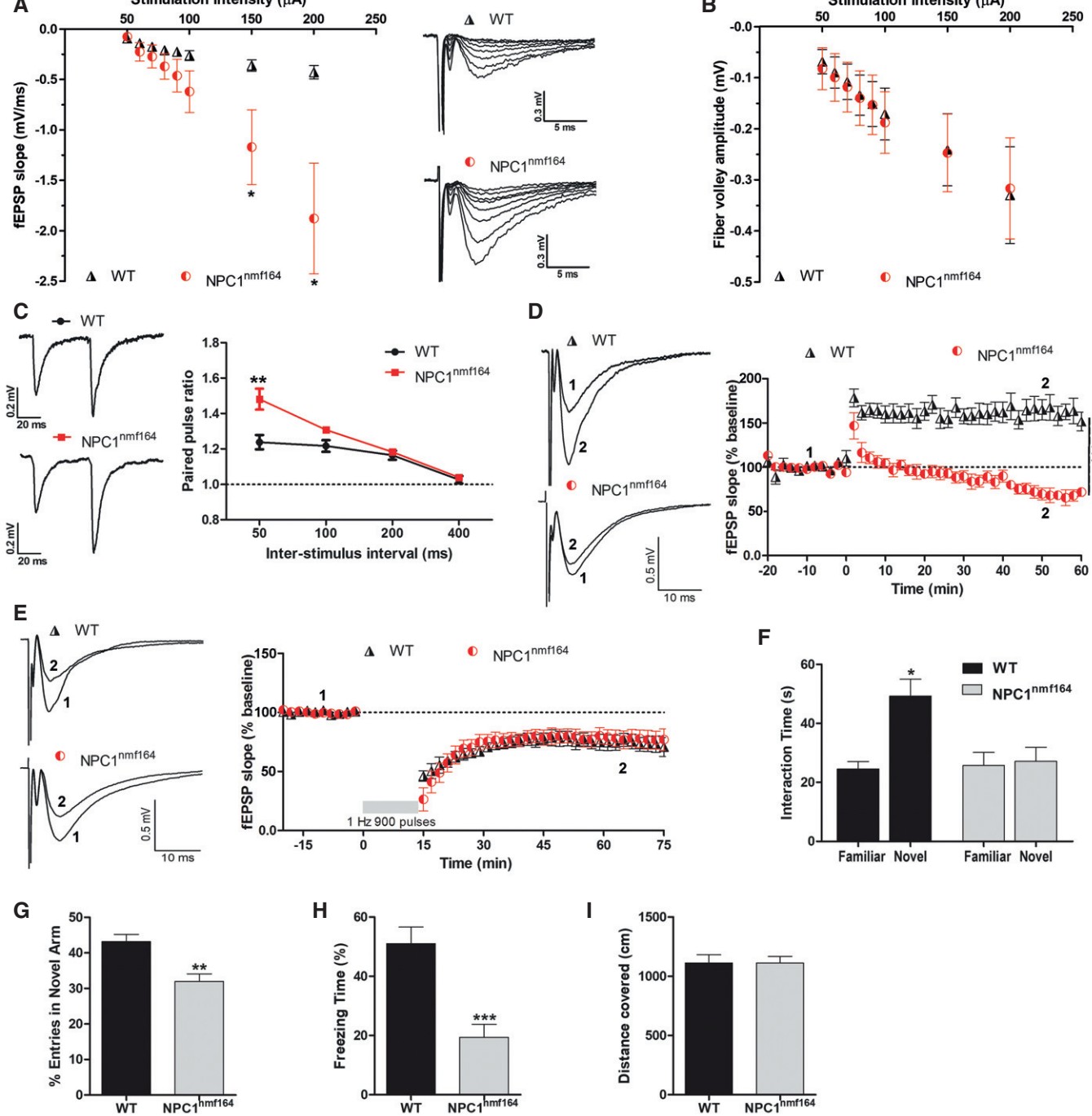

**Figure 3.**

◀

**Figure 3.  Synaptic function and behaviour in wt and *NPC1^nmf164* mice.**

The following properties of Schaffer collateral-CA1 synaptic responses (A–E) and behavioural tests (F–I) were recorded from wt and NPC1^nmf164 mice.

A   Basal synaptic transmission expressed as mean ± SEM EPSP slope ($n$ = 9 slices from 5 WT mice and $n$ = 8 slices from 5 NPC1^nmf164 mice, all mice 10 weeks old, unpaired Student's $t$-test, $*P_{150\ \mu A}$ = 0.0368, $*P_{200\ \mu A}$ = 0.0138).

B   Mean ± SEM fibre volley amplitude ($n$ = 10 slices from 5 WT mice and $n$ = 7 slices from 5 NPC1^nmf164 mice, all mice 10 week-old).

C   Mean ± SEM paired pulse facilitation ($n$ = 11 slices from 5 WT mice and $n$ = 10 slices from 5 NPC1^nmf164 mice, all mice 10 weeks old, unpaired Student's $t$-test, $**P$ = 0.0011).

D   LTP expressed as mean ± SEM percentage of EPSP slope over baseline ($n$ = 7 slices form 5 WT mice for WT and $n$ = 10 slices from 5 NPC1^nmf164 mice, all mice 10 weeks old, unpaired Student's $t$-test, $***P$ < 0.0001 in the last 10-min recording).

E   LTD expressed as mean ± SEM percentage of EPSP slope over baseline ($n$ = 7 slices from 5 WT mice and $n$ = 12 slices from 5 NPC1^nmf164 mice, all mice 10 weeks old).

F   Mean ± SEM exploration time of objects in novel and familiar location in the object placement recognition test ($n$ = 10 mice, 10 weeks old, 2-way ANOVA, $*P$ = 0.0167).

G   Mean ± SEM percentage of entry to the novel arm of the Y maze test ($n$ = 10 mice, 10 weeks old, unpaired Student's $t$-test, $**P$ = 0.0015).

H   Mean ± SEM percentage of freezing time in the contextual fear conditioning test ($n$ = 10 mice, 10 weeks old; unpaired Student's $t$-test, $***P$ = 0.0002).

I   Mean ± SEM distance covered during a 5-min period in the open-field test ($n$ = 10 mice, 10 weeks old).

Source data are available online for this figure.

## Mutant NPC1 fails to support cholesterol redistribution and CYP46A1 and AMPA receptor surface delivery during LTP

Recent evidence has shown that LTP-triggered cholesterol redistribution and reduction is necessary for the synaptic delivery of AMPA receptors [18], which in turn allows LTP progression. The hypothesis that NPC1 mediates LTP-triggered cholesterol changes was first tested by analysis in organotypic slice cultures from the hippocampus of wt and *NPC1^nmf164* mice. To maximize the number of synapses undergoing plasticity, cLTP was pharmacologically induced, resulting in synapse activation similar to physiological LTP [29–31]. Slices were subsequently centrifuged at 100,000 $g$ to separate light vesicular membranes in the supernatant and heavy membranes in the pellet enriched in plasma membrane. In agreement with the results reported by Brachet *et al* [18], cLTP induced a 30.4% reduction of cholesterol in the plasma membrane-enriched fraction of wt slices. In contrast, no significant change was observed in slices from *NPC1^nmf164* mice (Fig 4A). Cholesterol redistribution after LTP induction was further analysed by imaging experiments in primary cultures of hippocampal neurons from wt mice whose NPC1 expression was 64.2% reduced by a lentivirus carrying an NPC1 shRNA (Fig 4B). Cholesterol was labelled by transfection of neurons with a cDNA from the cholesterol-binding domain (D4) of perfringolysin O [32] fused to mCherry [33]. The specificity of mCherry-D4 for cholesterol was confirmed by the augmented signal observed in cells to which cholesterol was previously added complexed to methyl-β-cyclodextrin (Appendix Fig S7). In control neurons infected by lentivirus harbouring unspecific sh-scr, mCherry-D4 showed a diffuse plasma membrane-like distribution (Fig 4C). In contrast, in cells with reduced NPC1 levels (sh-NPC1), mCherry-D4 showed a punctate pattern compatible with vesicular fractions (Fig 4C). In sh-scr-expressing neurons, cLTP provoked an obvious redistribution of mCherry-D4 from the plasma membrane to light vesicular-like fractions, with no significant changes in the sh-NPC1-expressing neurons (Fig 4C).

Mobilization of the ER protein cholesterol hydroxylase CYP46A1 to the plasma membrane occurs upon excitatory stimulus by unknown means [21] and drives cholesterol reduction in LTP [18]. It has been proposed that an ER-derived carrier interacts with the plasma membrane allowing CYP46A1 to be exposed to the outside of the cell, thus releasing hydroxylated cholesterol. It has also been put forward that NPC1 moves cargos within and across the lysosomal membrane and could promote tethering between organelles. These suggestions prompted us to test the possibility that NPC1 facilitates CYP46A1 surface expression at synapses during LTP. Thus, we measured the presence of NPC1 and CYP46A1 in the surface of synaptosomes from wt and *NPC1^nmf164* mice by biotinylation assays in basal and cLTP conditions. cLTP-induced increment in NPC1 levels was accompanied by enhanced surface expression of the protein in wt but not in *NPC1^nmf164* synaptosomes (Fig 4D). Enhanced surface expression was also observed for CYP46A1 in the wt but not in the *NPC1^nmf164* synaptosomes (Fig 4E).

Cholesterol reduction is necessary for AMPA receptor surface delivery in LTP [18]. To test whether this was also prevented by NPC1 deficiency, total and surface immunocytochemical staining against an extracellular epitope of the AMPA receptor subunit GluA1 was performed in cultured neurons where NPC1 expression was silenced or not using lentivirus (Fig 4F). As expected, cLTP enhanced surface delivery of GluA1 (95.5% increase) in sh-scr-expressing neurons. In contrast, GluA1 surface delivery in response to cLTP was blocked in sh-NPC1 neurons (Fig 4F). Of note, basal surface levels of GluA1 were normal in sh-NPC1-expressing neurons.

## *In vitro* pharmacological activation of CYP46A1 restores cholesterol levels, LTP and GluA1 surface delivery in NPC1-deficient synapses

The above results indicate impaired cholesterol redistribution and CYP46A1 surface delivery during LTP and an increase in synaptic cholesterol levels when NPC1 is deficient. Since CYP46A1 is the enzyme responsible for cholesterol turnover in neurons [20], mediating cholesterol reduction during excitatory neurotransmission [19], we postulated that CYP46A1 activation might counteract cholesterol excess and LTP hindrance upon NPC1 deficiency. We first confirmed that both protein and mRNA levels of CYP46A1 were not significantly altered in total brain extracts and synaptosomes from *NPC1^nmf164* mice compared to wt mice (Fig 5A and B). This supported CYP46A1 pharmacological activation as a suitable strategy. To carry out this strategy, we used the anti-HIV medication efavirenz (EFV) as it promotes a robust stimulation of CYP46A1 [34]. Incubation with 20 µM EFV reduced cholesterol in synaptosomes from both wt and *NPC1^nmf164* mice (a 16.1 and 18.5%

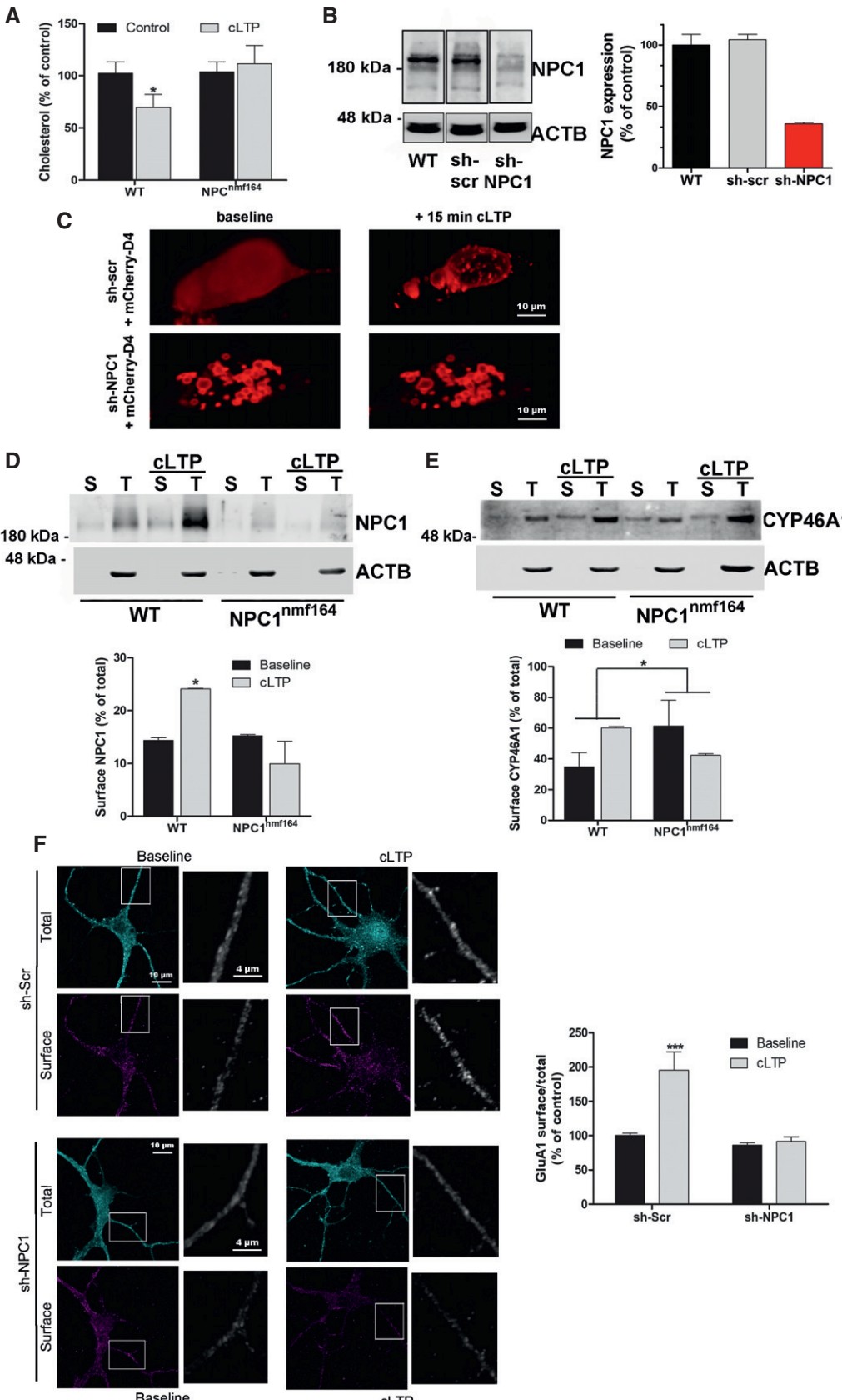

Figure 4.

**Figure 4. Cholesterol redistribution and CYP46A1 and GluA1 receptor surface delivery during LTP induction in wt and *NPC1^{nmf164}* mice.**

A   Mean $\pm$ SEM cholesterol level in plasma membrane fractions of hippocampal slice cultures from wt and *NPC1^{nmf164}* mice with or without cLTP induction ($n = 6$, unpaired Student's *t*-test, $*P_{wt} = 0.0402$).

B   Western blot of NPC1 and ACTB in extracts of cultured hippocampal neurons from wt mice transfected or not with sh-scr or sh-NPC1 RNAs. Black bars indicate that the sh-NPC1 lane was not consecutive to the others but belongs to the same Western blot. Graph shows the 2 biological replicates (mean $\pm$ SEM) of NPC1 level normalized to ACTB as a percentage of the wt, non-transfected controls.

C   Representative images of the cholesterol-binding probe mCherry-D4 before and after cLTP in cultured hippocampal neurons from wt mice transfected with sh-scr or sh-NPC1 RNAs. *Z*-stack images acquired using a Nikon A1R$^+$ confocal microscope were used to generate 3D images of the neurons.

D, E   Western blots against NPC1, CYP46A1 and actin-β (ACTB) in total extracts (T) and biotin–streptavidin immunoprecipitates (S) from synaptosomes from wt and *NPC1^{nmf164}* mice in which cLTP was induced or not. Graphs show mean $\pm$ SEM of the levels of biotinylated-surface NPC1 and CYP46A1 as a percentage of the total amount of each protein ($n = 3$ mice, 3 months old, 2-way ANOVA, $*P_{NPC1} = 0.0251$, $*P_{CYP46A1} = 0.0371$).

F   Representative images of GluA1 immunocytochemical staining before and after cLTP in non-permeabilized (surface) and permeabilized (total) cultured hippocampal neurons expressing sh-scr or sh-NPC1 RNAs. Graph shows mean $\pm$ SEM GluA1 surface staining with respect to total in cLTP conditions, as a percentage of the sh-scr baseline ($n = 20$ images per condition from two different experiments, 2-way ANOVA, $***P_{sh-scr} < 0.001$).

Source data are available online for this figure.

reduction, respectively) (Fig 5C). Though this treatment did not have a significant effect in slices from wt mice (Fig 5D–F), it restored synaptic functions such as basal synaptic transmission (Fig 5D) and LTP (Fig 5E), but not the paired pulse facilitation even after long time incubation (Fig 5F and Appendix Fig S8), in hippocampal slices from *NPC1^{nmf164}* mice. To test whether EFV treatment restored AMPA receptor delivery to the membrane upon LTP induction in NPC1-deficient neurons, we performed total and surface anti-GluA1 staining in sh-scr- and sh-NPC1-expressing neurons (Fig 6). cLTP in sh-scr neurons enhanced the surface delivery of GluA1 to a similar extent in EFV-treated and non-treated neurons (a 66.6 and 71.3% increase, respectively). Remarkably, EFV treatment counteracted the blockage of GluA1 surface delivery after cLTP in sh-NPC1 neurons, returning it to similar levels as in sh-scr neurons (Fig 6). EFV treatment did not alter basal surface levels of GluA1 in either sh-scr or sh-NPC1 neurons (Fig 6) or NPC1 levels and distribution in wt neurons (Appendix Fig S9).

**Oral treatment with EFV prevents cholesterol accumulation and memory impairment, and extends life span in NPC1^{nmf164} mice**

The positive effects of the *in vitro* EFV treatment on synaptic cholesterol levels and plasticity encouraged the *in vivo* testing of this drug. EFV has the ability to cross the blood–brain barrier and promotes brain cholesterol turnover in mice at doses 300 times lower than those used for HIV patients [34]. We administered this dose (0.09 mg/kg/day) for 8 weeks in the drinking water of 1.5-month-old wt and *NPC1^{nmf164}* mice. The CYP46A1 metabolite 24(S)-hydroxycholesterol was measured in the plasma after 6 and 8 weeks of treatment, reaching similar levels in non-treated wt and *NPC1^{nmf164}* mice. In wt mice, EFV treatment elicited a slight increase in 24(S)-hydroxycholesterol levels only at 6 weeks of treatment, whereas its levels were drastically increased in EFV-treated *NPC1^{nmf164}* mice, both at 6 and 8 weeks of treatment (a 3.5- and 2.2-fold increase, respectively) (Fig 7A). Weight gain, which was impaired in the *NPC1^{nmf164}* compared to the wt mice, was improved by the EFV treatment (Fig 7B). Functional effects in mice were determined by different behavioural tests. EFV treatment did not have significant effects on the behaviour of wt mice (Fig 7C–F). However, in *NPC1^{nmf164}* mice, EFV increased interest in moved objects (object placement recognition tests) and enhanced the number of entries into the novel arm in Y maze tests, indicative of improved hippocampal spatial learning and memory (Fig 7C and D). EFV rescued contextual (Fig 7E) and cued (Fig 7F) learning and memory in the fear conditioning paradigm in *NPC1^{nmf164}* mice. EFV treatment also improved motor abilities in the *NPC1^{nmf164}* mice. This could be explained by the prevention of Purkinje cell death in the cerebellum (Appendix Fig S10).

Synaptosomes were isolated from the mice to monitor the effects on synaptic cholesterol levels. EFV reduced the levels of

**Figure 5. Effects of CYP46A1 activation by EFV on synaptic cholesterol redistribution and plasticity in wt and *NPC1^{nmf164}* neurons.**

A   Western blots against CYP46A1 and ACTB in extracts of total brain and synaptosomes from wt and *NPC1^{nmf164}* mice as a percentage of wt values. Graphs show 3 biological replicates (mean $\pm$ SEM) of CYP46A1 level normalized to ACTB in arbitrary units.

B   Quantitative PCR of *Cyp46* of total brain and synaptosomes from wt and *NPC1^{nmf164}* mice. Graphs show 3 biological replicates (mean $\pm$ SEM).

C   Mean $\pm$ SEM cholesterol levels in EFV-treated or non-treated synaptosomes from wt ($n = 5$ mice, 14 weeks old, unpaired Student's *t*-test, $*P = 0.0233$) and *NPC1^{nmf164}* ($n = 4$ mice, 14 weeks old, unpaired Student's *t*-test, $*P = 0.0188$) mice. EFV values are expressed as percentage of their corresponding non-treated controls in wt and *NPC1^{nmf164}* conditions.

D   Basal synaptic transmission in EFV-treated and non-treated hippocampal slices from wt and NPC1^{nmf164} mice expressed as mean $\pm$ SEM EPSP slope ($n = 9$ slices from 5 WT mice, $n = 11$ slices EFV-treated from 5 WT mice, $n = 8$ slices from 5 NPC1^{nmf164} mice and $n = 13$ slices EFV-treated from 5 NPC1^{nmf164} mice, all mice 14 weeks old, 2-way ANOVA, $*P = 0.0117$).

E   LTP in EFV-treated and non-treated hippocampal slices from wt and NPC1^{nmf164} mice expressed as mean $\pm$ SEM percentage of EPSP slope over baseline ($n = 7$ slices from 5 WT mice, $n = 9$ slices EFV-treated from 5 WT mice, $n = 10$ slices from 5 NPC1^{nmf164} mice and $n = 11$ slices EFV-treated from 5 NPC1^{nmf164} mice, all mice 14 weeks old, 2-way ANOVA, $***P < 0.0001$).

F   Mean $\pm$ SEM paired pulse facilitation in EFV-treated and non-treated hippocampal slices from wt and NPC1^{nmf164} mice ($n = 11$ slices from 5 WT mice, $n = 9$ slices EFV-treated from 5 WT mice, $n = 10$ slices from 5 NPC1^{nmf164} mice and $n = 11$ slices EFV-treated from 5 NPC1^{nmf164} mice, all mice 14 weeks old, 2-way ANOVA, $**P_{NPC1nmf164} = 0.0027$, $*P_{NPC1nmf164+EFV} = 0.0229$).

Source data are available online for this figure.

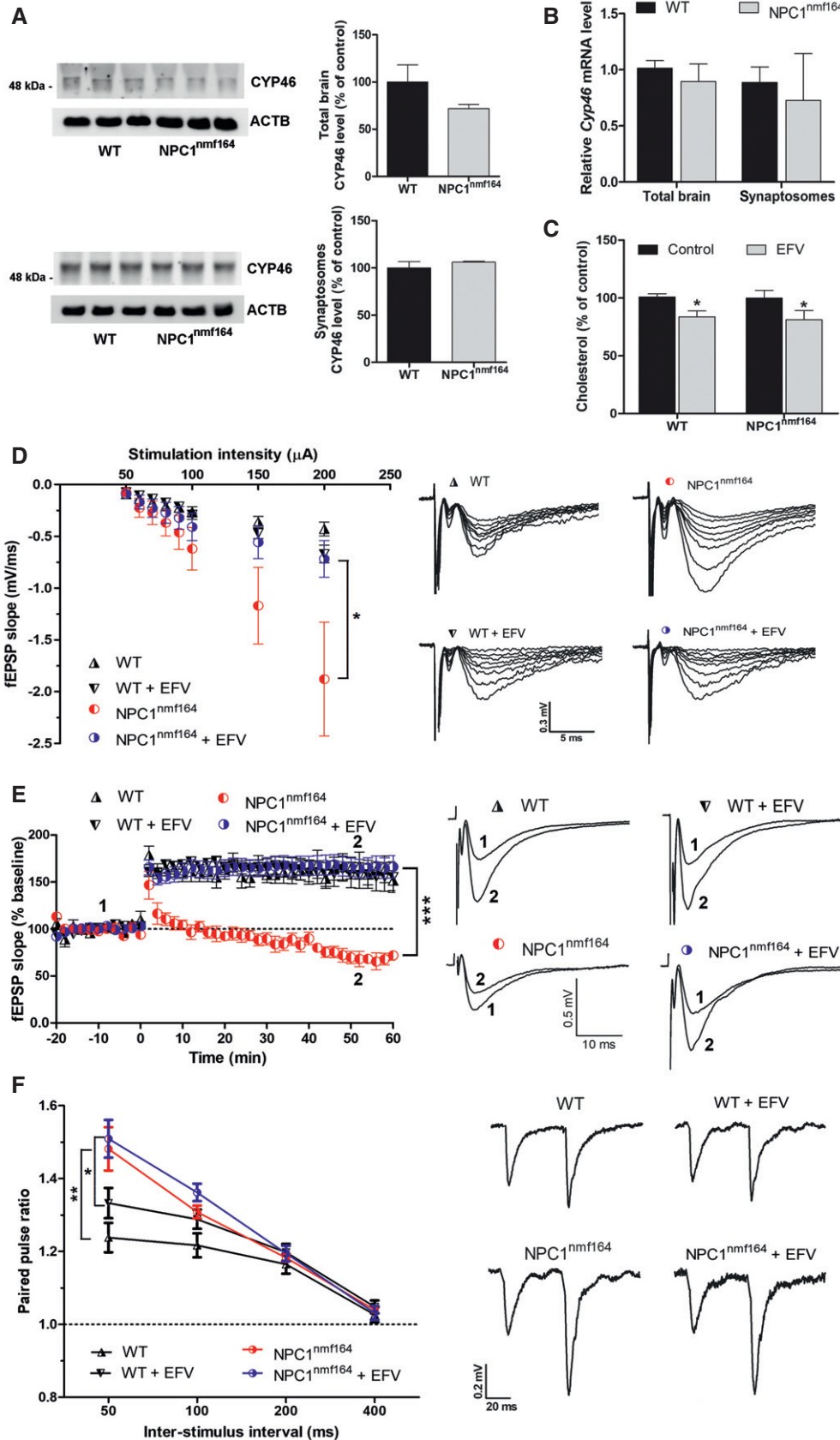

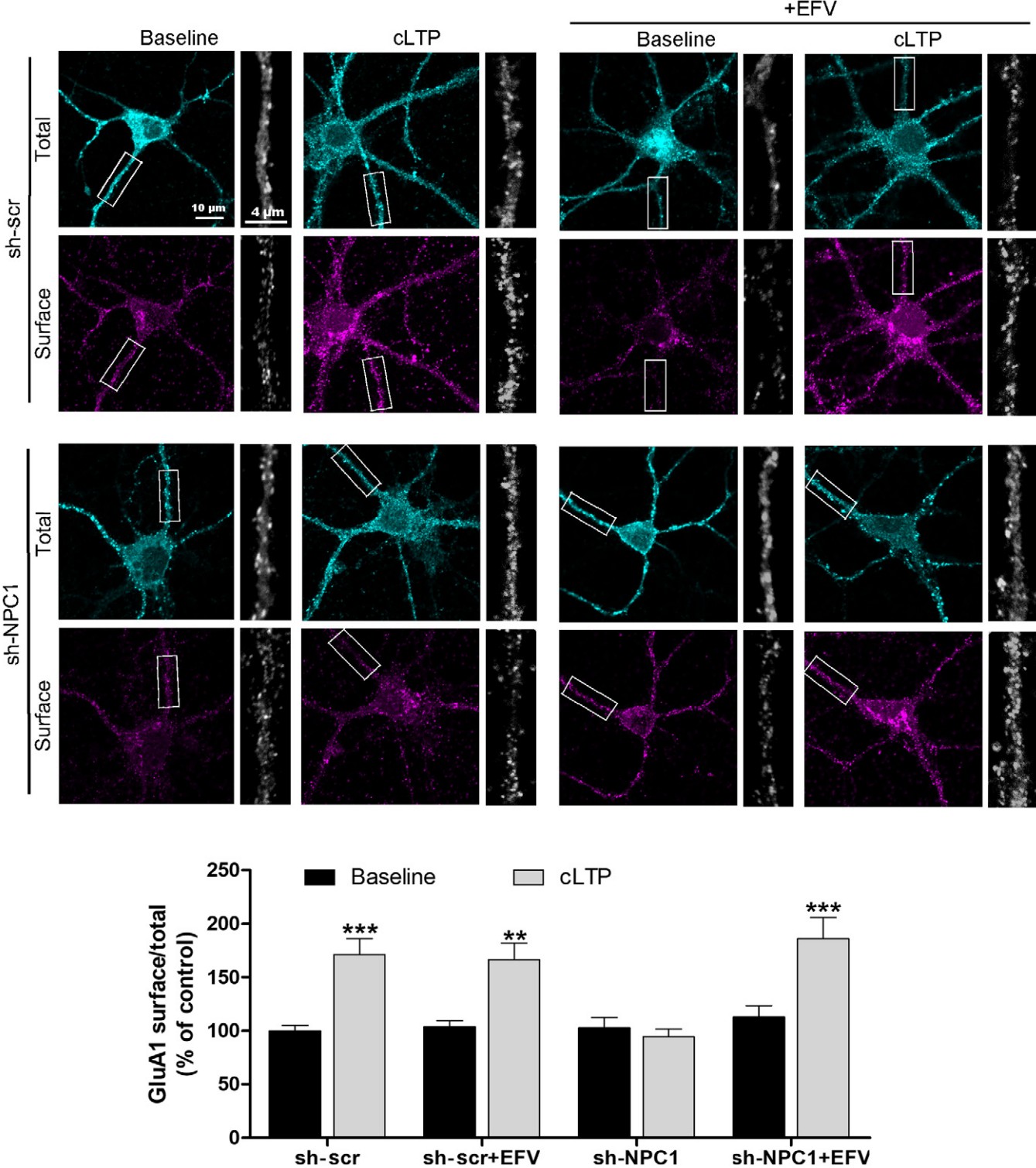

**Figure 6. Effects of CYP46A1 activation by EFV on cLTP-induced GluA1 surface delivery in wt- and NPC1-deficient conditions.**

Representative images of GluA1 immunocytochemical staining before and after cLTP induction in non-permeabilized (surface) and permeabilized (total) cultured hippocampal neurons expressing sh-scr or sh-NPC1 RNAs and treated or not with EFV. Graph shows the mean $\pm$ SEM GluA1 surface staining with respect to total after cLTP, as a percentage of the sh-scr baseline ($n = 20$ images per condition from 2 separate experiments, 2-way ANOVA, ***$P_{sh-scr} < 0.001$, **$P_{sh-scr+EFV} < 0.01$, ***$P_{sh-NPC1+EFV} < 0.001$).

cholesterol in synaptosomes of *NPC1^nmf164* and wt mice by 38.1 and 17.2%, respectively (Fig 7G), as determined by enzymatic assays. In addition, a cholesterol-binding fluorescent antibiotic (filipin) was used to examine cholesterol levels in hippocampal tissue. Since accumulation of this lipid in lysosomes is a hallmark of NPC1 deficiency, we performed co-staining with the lysosomal marker LAMP1. While filipin staining in wt mice was barely detectable regardless of EFV treatment, a strong signal was evident in the hippocampus of non-treated *NPC1^nmf164* mice, consistent with increased levels of cholesterol (Fig 7H). EFV significantly reduced filipin staining in the hippocampus of *NPC1^nmf164* mice by

33.9% (Fig 7H). LAMP1 staining in the non-treated *NPC1^nmf164* mice showed the accumulation of enlarged lysosomes typical of NPC1 deficiency, a phenotype that was not evident in the hippocampus after EFV treatment (Fig 7H). Consistent with the brain specificity of CYP46A1, we did not detect changes in the liver on the levels of cholesterol, measured by filipin staining and enzymatic assays, or inflammation, measured by F4/80 macrophage staining (Appendix Fig S11). Given the positive effects of EFV on behaviour and brain cholesterol accumulation after 8 weeks of treatment, we extended EFV administration in a group of wt and *NPC1^nmf164* mice to determine the effects on survival.

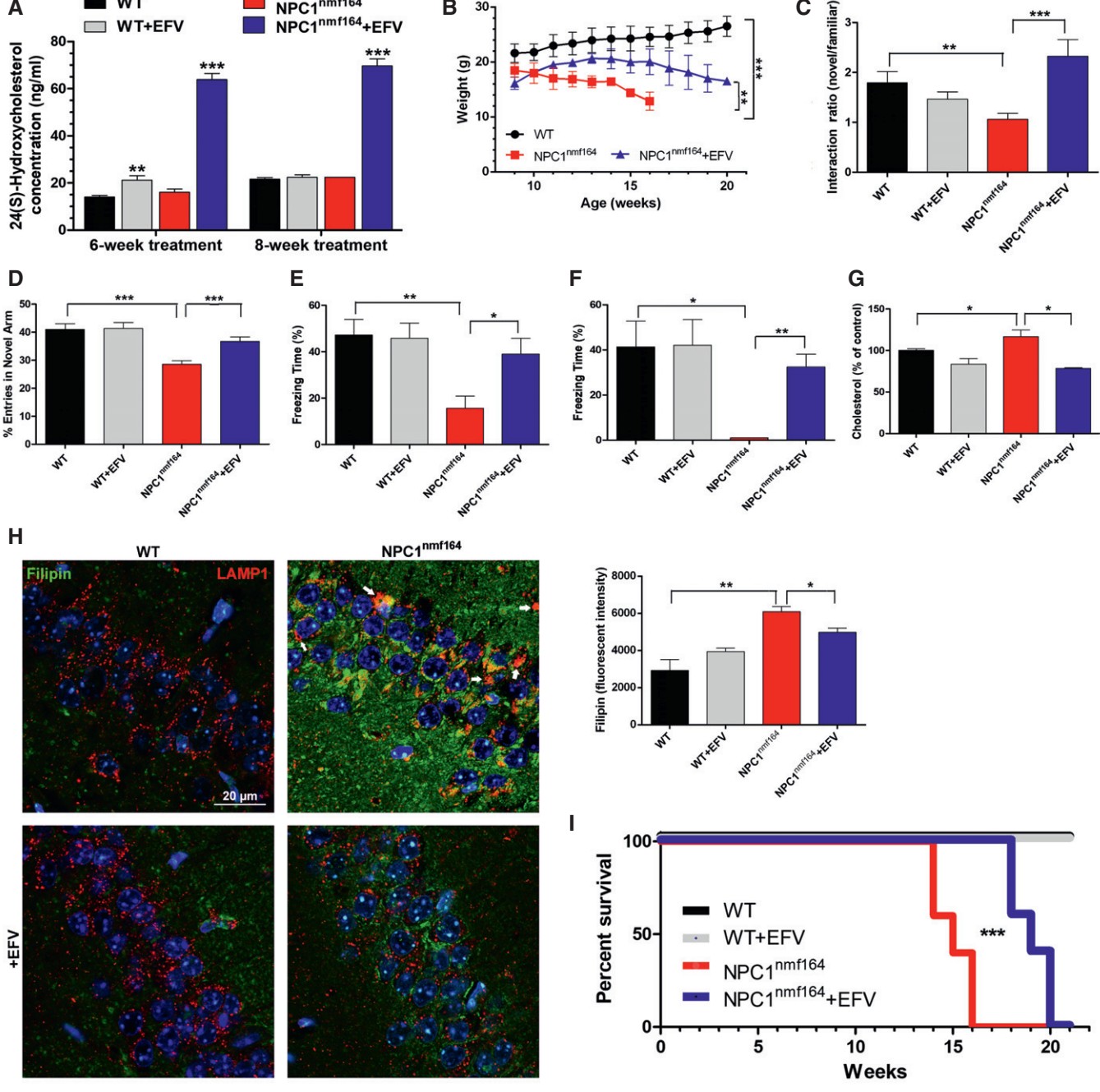

**Figure 7.**

◄ **Figure 7. Effects of EFV treatment *in vivo*.** The following analyses and behavioural tests were assessed in wt and $NPC1^{nmf164}$ mice treated or not with 0.09 mg/kg/day EFV.

A Mean ± SEM 24(S)-hydroxycholesterol plasma level after 6 and 8 weeks of oral EFV treatment in wt ($n$ = 6 mice, 14 weeks old and 16 weeks old, 2-way ANOVA, **$P_{6\ weeks}$ = 0.0023) and $NPC1^{nmf164}$ ($n$ = 5 mice, 14 weeks old and 16 weeks old, 2-way ANOVA, ***$P_{6\ weeks}$ < 0.0001, ***$P_{8\ weeks}$ < 0.0001) mice.

B Mean ± SEM body weight in grams in wt and $NPC1^{nmf164}$ mice treated or not with EFV ($n$ = 4 mice, unpaired Student's $t$ test, ***$P_{NPC1^{nmf164}}$ < 0.0001, **$P_{NPC1^{nmf164}+EFV}$ = 0.0048).

C Mean ± SEM discrimination index as a normalized ratio of time spent with novel and familiar objects in the object placement recognition test ($n$ = 6 mice, 10 weeks old, 2-way ANOVA, **$P_{WT/NPC1nmf164}$ = 0.0065, ***$P_{NPC1nmf164/NPC1nmf164+EFV}$ = 0.0008).

D Mean ± SEM percentage of entries in the novel arm of the Y maze test ($n$ = 10 mice, 10 weeks old, 2-way ANOVA, ***$P$ < 0.0001).

E Mean ± SEM percentage of freezing time in the contextual fear conditioning test ($n$ = 10 mice, 10 weeks old, 2-way ANOVA, **$P_{NPC1}$ < 0.0025, *$P_{NPC1+EFV}$ < 0.0181).

F Mean ± SEM percentage of freezing time in the cued fear conditioning test ($n$ = 10 mice, 10 weeks old, 2-way ANOVA, *$P_{NPC1}$ < 0.0453, **$P_{NPC1+EFV}$ < 0.0054).

G Mean ± SEM cholesterol level in synaptosomes of wt and $NPC1^{nmf164}$ mice treated or not with EFV expressed as percentage of the wt non-treated samples ($n$ = 4 mice, 14 week-old, 2-way ANOVA, *$P_{NPC1}$ < 0.0476, *$P_{NPC1+EFV}$ < 0.0118).

H Representative fluorescence images of the CA1 hippocampal region from wt and $NPC1^{nmf164}$ mice treated or not with EFV and stained with filipin and an antibody against LAMP1. White arrows indicate lysosome enlargement in the hippocampus of non-treated $NPC1^{nmf164}$ mice. Graph to the right shows mean ± SEM fluorescence intensity associated with filipin per area in arbitrary units ($n$ = 4 mice, 14 weeks old, unpaired Student's $t$-test, **$P_{NPC1^{nmf164}}$ = 0.0011, *$P_{NPC1^{nmf164}+EFV}$ = 0.0273).

I Survival graph for wt and NPC1$^{nmf164}$ mice treated or not with EFV ($n$ = 5 mice, 2-way ANOVA, ***$P$ < 0.0001).

All non-treated $NPC1^{nmf164}$ mice died before 16 weeks of age, with a mean survival of 100.2 ± 2.2 days. None of the EFV-treated NPC1$^{nmf164}$ mice died before 16 weeks of age, with a mean survival time of 129.4 ± 2.7 days (Fig 7I).

## Discussion

NPC1 is known for its contribution to cholesterol transport across the endolysosomal compartment of all cells, facilitating the egress of cholesterol [5,6,35–37]. Despite its ubiquitous presence, the impact of NPC1 deficiency is particularly significant in neurons. In fact, cognitive and psychiatric alterations are frequent consequences of NPC1 mutations in NPC patients [7,38]. Our results unveiling a key role for NPC1 in synaptic function contribute to explain neuronal vulnerability to NPC1 deficiency. Using mice bearing a mutation in the NPC1 gene that is similar to a common one in NPC patients, we demonstrated that NPC1 defects have a profound impact on synapses. Besides the morphological alterations in synaptic vesicles, which are in agreement with a previous electron microscopy study in *Npc1*-null mice [10], we have unveiled the relative enrichment of the protein at the postsynapse and its contribution to synaptic plasticity. NPC1 mediates the cholesterol redistribution and CYP46A1 and AMPA receptor dynamism necessary for LTP, which drives the postsynaptic plasticity required for learning and memory. Impairment of these events by NPC1 mutation may explain the progressive cognitive deficits in NPC patients. Since increasing evidence links synaptic plasticity not only to cognition but also to mood regulation [39,40], these deficits could also account for psychiatric alterations that characterize NPC patients. Our work encourages further research to address this issue.

Impaired LTD and LTP had been reported in the cerebellum and hippocampus, respectively, of *Npc1*-null mice [16,41]. While we confirmed LTP impairment in the hippocampus of the $NPC1^{nmf164}$ mice, we did not observe changes in the hippocampal LTD. The different brain area and mouse model studied may account for this discrepancy. Reduced levels of adenosine were suggested as the cause for the LTD and LTP alterations found in the *Npc1*-null mice. Independently of a deficiency in extracellular factors, our results highlight the intrinsic failure of a synaptic cholesterol-dependent

mechanism impairing LTP in NPC1 mutant mice. Very recently, a link between lysosomal cholesterol and neuronal firing patterns has been revealed in neurons upon NPC1 pharmacological inhibition or genetic mutation [42]. Alterations in lysosomal cholesterol egress transcriptionally upregulate the ABCA1 transporter, leading to reduced levels of phosphoinositides at the plasma membrane, which in turn decrease a voltage-gated potassium channel and enhance excitability. Enhanced excitability would be consistent with the increased basal synaptic transmission we observed in hippocampal neurons from $NPC1^{nmf164}$ mice. Altogether, these results emphasize the involvement of lipids and their different ways to fine-tune neuronal function.

Our results highlight the relevance of cholesterol dynamics in LTP and cognitive function, and the cooperative action of two cholesterol-related proteins, NPC1 and CYP46A1, for its efficient progression. The observation that, in mutant NPC1 synapses, LTP-induced CYP46A1 surface expression and cholesterol reduction are impaired suggests a role for NPC1 in the mobilization of CYP46A1 to the plasma membrane where this enzyme would contribute to cholesterol release enabling AMPA receptor surface delivery necessary for LTP progression (Fig 8A). LTP-induced local translation of NPC1 and the proposed capacity of this protein to move cargos and build contact sites between membrane organelles could contribute to these effects. While further work is required to determine the exact mechanism, we propose that NPC1 could reach the plasma membrane from the endolysosomal compartment and act as a tether facilitating ER-plasma membrane contact and CYP46A1 surface delivery. Pharmacological activation of CYP46A1 by EFV would compensate the low surface levels of the enzyme in the $NPC1^{nmf164}$ mice, restoring cholesterol reduction, AMPA receptor delivery and thus synaptic plasticity and cognitive abilities (Fig 8B and C). We do not rule out that in addition, and due to its cholesterol-binding abilities, NPC1 directly mediates LTP-induced cholesterol redistribution between vesicular and plasma membrane fractions. NPC2, which we also find in synaptosomes (Appendix Fig S12), could help NPC1 in this task. Our results also point to the enhanced degradation of NPC1 due to the D1005G mutation, rather than to impaired transport, as the cause for the drastic reduction of the protein levels in the synapses of $NPC1^{nmf164}$ mice. This finding

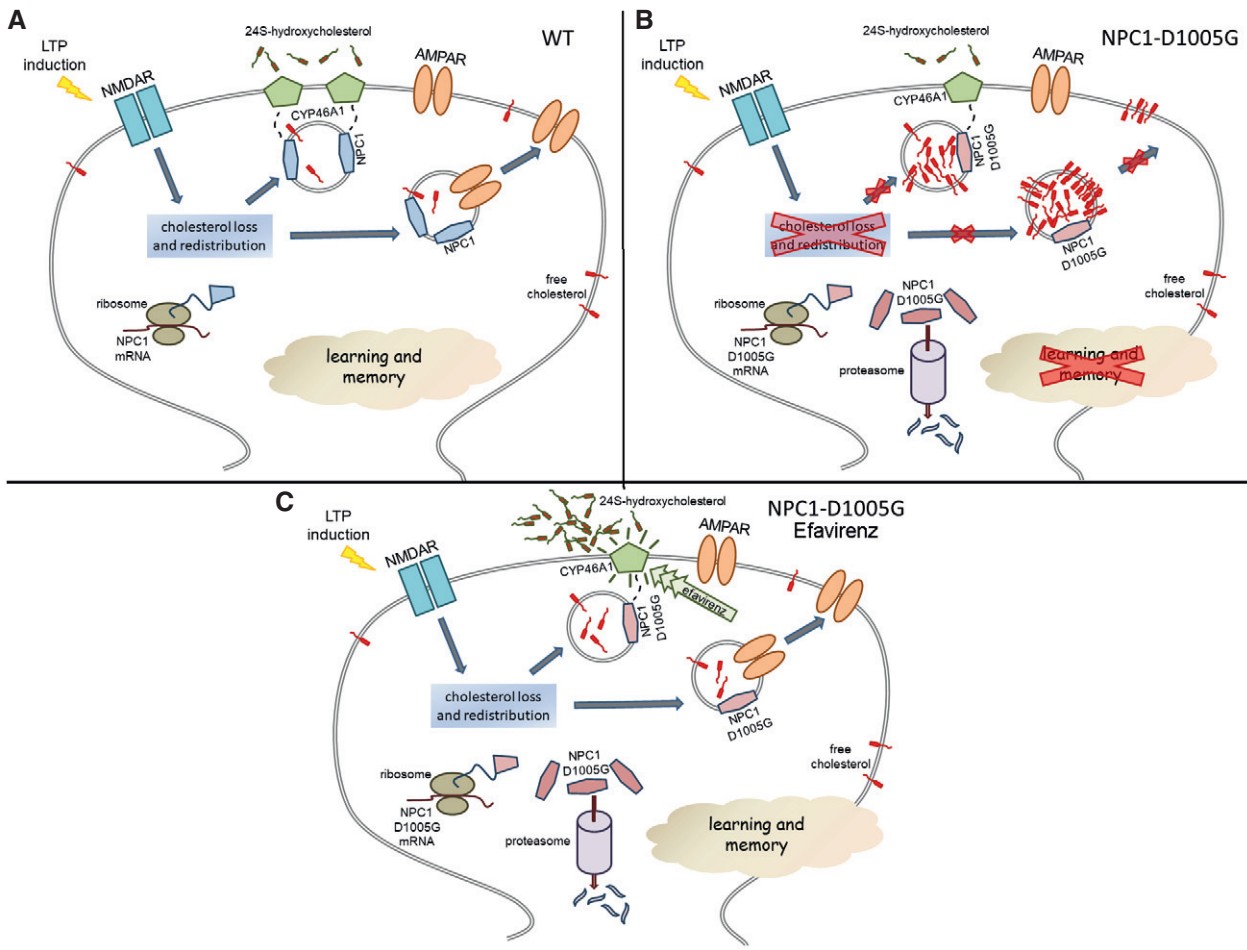

**Figure 8. Model for NPC1 function in synapses and its pharmacological rescue in *NPC1^nmf164* mice.**

A–C With this model, we propose that in the wt situation, LTP induction promotes the synthesis of NPC1, which enhances the surface delivery of CYP46A1. While the exact mechanism mediating this effect remains to be determined, we propose NPC1 could reach the plasma membrane from the endolysosomal compartment and act as a tether between ER and the plasma membrane facilitating the surface delivery of CYP46A1. This allows cholesterol hydroxylation and release that is necessary for AMPAR plasma membrane delivery, LTP progression and thus learning and memory. The NPC1 mutation D1005G enhances the protein degradation in synapses preventing LTP-induced NPC1 increase. This hampers CYP46A1 surface expression and cholesterol reduction resulting in less AMPAR delivery to the plasma membrane and impaired learning and memory. The use of EFV activates CYP46A1, restoring cholesterol reduction, AMPAR surface delivery, LTP and learning and memory abilities.

strengthens NPC1 chaperones as suitable therapeutic strategy for the cognitive problems in NPC patients bearing this kind of mutation.

There is currently no cure for NPC; the only treatment available is administration of the synthetic iminosugar miglustat, which inhibits glucosylceramide synthase, reducing ganglioside build-up [43]. Miglustat treatment delays the neurological progression of NPC patients but does not cure the disease, and has important side effects. Finding new therapeutic strategies for NPC is therefore an urgent need. Several compounds are currently being considered. Histone deacetylase inhibitors reduced cholesterol accumulation and increased NPC1 levels in NPC cellular and animal models [44,45]. Cholesterol reduction and moderate improvement in life span were achieved in NPC animal models by either activation of the heat-shock protein HSP70 with the drug arimoclomol [46] or administration of the cholesterol-sequestering drug β-cyclodextrin

[36,47–49]. Clinical trials with these compounds have been approved for NPC, but their broad targets, potential toxicity and/or invasive administration due to poor penetration through the blood–brain barrier raise concern about their application in patients. Although having visceral involvement, NPC is, above all, a neurological disorder, and therefore, brain pathology is the major target. We here demonstrate the benefits of oral treatment with the CYP46A1-activating compound EFV for brain pathology in a mouse model for NPC. EFV is an approved medication for human use, already prescribed for the treatment of HIV [34]. While having a good tolerability, neurological side effects such as depression or cognitive deficits have been described in certain HIV patients chronically treated with EFV [50]. We believe our data help to understand these effects. EFV would enhance CYP46A1 activity reducing cholesterol levels in brains that, different from the NPC patients, do not have an excess of this lipid. Since cholesterol is necessary for LTP,

its abnormal reduction could impair synaptic plasticity contributing to the neurological alterations described in HIV patients chronically treated with EFV. Importantly, the dose that showed efficacy in the NPC mouse model is much lower than that used in HIV patients. Altogether, the evidence argues in favour of the safety and clinical translation of chronic EFV administration as a therapeutic approach for NPC. Even more, our results showed a notable increase in plasma levels of 24-hydroxycholesterol in the EFV-treated *NPC1^nmf164* mice, providing a suitable biomarker to follow the treatment. The benefits of EFV treatment not only concern synaptic function and cognitive performance, but we also observed improvement in the lysosomal phenotype in brain tissue, and extension of almost 30% in life span in the mouse model. It is important to keep in mind that CYP46A1 is brain-specific, and therefore, it is unlikely that EFV treatment would address peripheral pathology. This is supported by the lack of effect on cholesterol levels and inflammation we observed in the liver of EFV-treated *NPC1^nmf164* mice. This may explain the limited, though significant, extension of survival and encourages combined treatment using EFV with the aforementioned compounds currently in clinical trials that can target peripheral pathology in animal models. Testing these combined therapies is a task for the future that may open exciting new perspectives for the treatment of NPC.

# Materials and Methods

### Mice and ethics statement

A breeding colony of *NPC1^nmf164* mice carrying a D1005G mutation in *Npc1* [22] was established at the CBMSO from C57BL/6J-Npc1nmf164/J heterozygous mice purchased from the Jackson laboratories. Male and female wt and homozygous NPC1^nmf164 littermates were identified by PCR and were randomly assigned to experimental groups. The mice were kept in a 12-h light/dark cycle in SPF (specific pathogen free) room. Age of mice in each experiment is indicated in the Figure legends. No gender-dependent differences were observed in any of the results. Experiments were conducted according to the ARRIVE guidelines. Review boards at the CBMSO, the Spanish Research Council and the Comunidad de Madrid approved the procedures involving mice (PROEX 175/17), which were performed in accordance with specific European Union guidelines (Directive 2010/63/EU).

### Antibodies

Antibodies against the following proteins were used for Western blots and immunofluorescence: NPC1 (rabbit polyclonal, Novus Biologicals, UK, NB 400-148; rabbit monoclonal, Abcam, #ab134113), PSD-95 (mouse monoclonal, NeuroMab, Ca, USA, 75-028), synaptophysin-1 (polyclonal guinea pig, Synaptic Systems, Germany, 101004), CYP46A1 (rabbit polyclonal, Proteintech, USA, 12486-1-AP), GluR1 C terminus (rabbit polyclonal (Abcam, ab31232), GluR1N terminus (mouse monoclonal, Merck-Millipore, MAB2263), LAMP1 (rat monoclonal, DSHB) and anti-β-actin (mouse monoclonal Sigma-Aldrich, A2228). Alexa Fluor or HRP-conjugated goat anti-rabbit, rabbit anti-mouse and donkey anti-rat antibodies (Life Technologies) were used as secondary antibodies.

### Neuronal cultures

Primary cultures of hippocampal neurons were prepared from mouse day-18 (E18) embryos as described [51]. Neurons were cultured in Neurobasal medium (Thermo Fisher Scientific, 21103–049) supplemented with B27 (Thermo Fisher Scientific, 17504044) and GlutaMAX (Thermo Fisher Scientific, 35050061). At 7 days *in vitro* (DIV), the culture medium was replaced with medium without GlutaMAX and used for experiments at 14 DIV.

### Organotypic slice cultures

Organotypic hippocampal slice cultures were prepared as described [52]. Hippocampi were dissected from postnatal day 7 mice, and 400 μm slices were prepared. The slices were maintained in medium containing 20% horse serum, 1 mM L-glutamine, 1 mM $CaCl_2$, 2 mM $MgSO_4$, 1 mg/l insulin, 0.0012% ascorbic acid, 30 mM HEPES, 13 mM D-glucose and 5.2 mM $NaHCO_3$, at 35.5°C and used after 7 DIV.

### Isolation of synaptosomes

Mouse brains were homogenized in 0.32 mM sucrose, 1 mM EDTA, 1 mg/ml BSA and 5 mM HEPES (pH 7.4), and centrifuged at 3,000 ×*g* for 12 min at 4°C. The supernatant was further centrifuged for 12 min at 14,000 ×*g* at 4°C, and the pellet obtained was resuspended in Krebs–Ringer buffer (140 mM NaCl, 5 mM KCl, 5 mM glucose, 1 mM EDTA, 10 mM HEPES, pH 7.4) and mixed with Percoll (45% v/v). The resulting solution was centrifuged for 2 min at 16,000 ×*g* at 4°C. The synaptosomes were collected from the surface with a syringe and resuspended in Krebs–Ringer buffer, followed by a centrifugation at 16,000 ×*g* for 30 s at 4°C. Synaptosomes were obtained in the pellet that was resuspended in HEPES–Krebs buffer (147 mM NaCl, 3 mM KCl, 10 mM glucose, 2 mM $MgSO_4$, 2 mM $CaCl_2$, 20 mM HEPES, pH 7.4).

### Electron microscopy and immunogold labelling

NPC1^nmf164 mice and wt littermates ($n = 3$ per genotype) were intracardially perfused with phosphate-buffered saline (PBS) and fixative (4% paraformaldehyde [PFA] and 2% glutaraldehyde [GTA] in 0.1 M phosphate buffer, pH 7.4, or 2% PFA and 0.2% GTA in the same buffer for immunogold labelling). Brains were fixed in 4% PFA overnight (ON) and sectioned in 200-μm-thick slices (300-μm-thick for immunogold labelling). Hippocampal sections were postfixed in 1% osmium tetroxide (in 0.1 M PBS, pH 7.4), dehydrated in ethanol and embedded in Epon (TAAB 812 Resin, TAAB Laboratories). Serial ultrathin sections of the CA1 region were stained with uranyl acetate and lead citrate. The samples for immunogold labelling were quenched for free aldehydes with 0.05 M $NH_4Cl$. Sections were cryoprotected in glycerol, and freeze-substitution was carried out at −90°C in methanol containing 0.5% uranyl acetate for 80 h. 70-nm ultrathin CA1 hippocampal sections were incubated with the NPC1 primary antibody (1:20), followed by a secondary antibody coupled to 15-nm gold particles. Sections were stained with uranyl acetate and lead citrate and examined with a transmission electron microscope (JEM1010; JEOL, Akishima, Tokyo, Japan). Synapses from CA1 were photographed at

a 20,000× magnification with a CMOS 4 k TemCam-F416 camera (TVIPS, Gauting, Germany). Images were quantified using ImageJ software (National Institutes of Health, Bethesda, MD, USA), and SV density was calculated as the number of vesicles per $\mu m^2$ within 10 nm of the presynaptic membrane and not more than 300 nm from the active zone border.

## Immunocytochemistry

Neurons were fixed with 4% PFA and 4% sucrose in PBS at RT. Non-specific binding was blocked with 0.2% gelatine and 1% BSA in PBS. For labelling surface GluA1, neurons were incubated for 2 h with an antibody against the GluA1N terminus. Total GluA1 was then detected after a permeabilization step (30-min incubation with 0.1% Triton X-100, 0.2% gelatine and 1% BSA in PBS) with an antibody against the GluR1 C terminus. Corresponding secondary antibodies conjugated to Alexa Fluor 555 and Alexa Fluor 647 (Life Technologies) were incubated for 1 h. Coverslips were mounted in Permount (Life Technologies), and cells were imaged using an inverted confocal microscope (LSM800; Carl Zeiss). Quantification was performed using ImageJ software, and regions of interest corresponding to individual dendrites were selected on the total GluA1 channel. Thus, image quantification was blind with respect to the surface GluA1 channel.

## Chemical induction of LTP

cLTP on synaptosomes was induced as described [23,24]. Freshly prepared synaptosomes (180 μl containing ~900 μg protein) were incubated with cLTP solution (HEPES–Krebs buffer without $MgSO_4$) supplemented with 500 μM glycine, 0.01 mM strychnine and 0.2 mM picrotoxin for 20 min at 37°C. After glycine treatment, synaptosomes were depolarized with 100 μl of cLTP solution with 0.001 mM strychnine, 0.02 mM picrotoxin and 53 mM KCl and incubated for 60 min. As control, synaptosome samples were maintained in HEPES–Krebs buffer (37°C, 80 min) and treated with identical volumes of HEPES–Krebs buffer instead of glycine or KCl.

cLTP on hippocampal organotypic slice and neuronal cultures was induced as described [18]. Hippocampal slices were incubated in artificial cerebrospinal fluid (ACSF; 119 mM NaCl, 2.5 mM KCl, 1 mM $NaH_2PO_4$, 11 mM D-glucose, 26 mM $NaHCO_3$, 1.25 mM $MgCl_2$ and 2.5 mM $CaCl_2$) saturated with 95% $O_2$/5% $CO_2$ for 5 min at RT. LTP was induced with 0.1 μM rolipram, 50 μM forskolin and 100 μM picrotoxin in ACSF lacking $MgCl_2$ for 15 min at RT. Slices were washed in cold PBS, homogenized in MES buffer (25 mM MES, 2 mM EDTA and a cocktail of protease and phosphatase inhibitors [Roche]) and centrifuged at 100,000 ×$g$ for 2 h at 4°C to separate the plasma membrane (pellet, resuspended in 0.1% SDS in PBS) and light vesicular membrane fraction (supernatant). In neuronal cultures, the medium was replaced with ACSF at 17 DIV. For the mCherry-D4 cholesterol redistribution, baseline images were acquired after 15 min when the solution was switched to ACSF containing 0.1 μM rolipram, 50 μM forskolin, and 100 μM picrotoxin without $MgCl_2$. After a 15-min treatment, fluorescence was collected as Z-stacks using a Nikon A1R$^+$ confocal microscope. For GluA1 receptor recycling, cLTP was induced as previously described followed by cell fixation. Images were taken with a confocal LSM800 microscope (Carl Zeiss AG, Germany).

## Surface biotinylation

Synaptosomes were washed with ice-cold PBS and incubated with 2 mg/ml of EZ-Link Sulfo-NHS-Biotin (#21217, Thermo Scientific) for 1 h at 4°C. The excess of biotin was quenched by washing three times with 100 mM L-Lys in PBS. Synaptosomes were lysed in RIPA buffer supplemented with protease inhibitors. 40 μl of Pierce Streptavidin Plus UltraLink Resin (#53116, Thermo Scientific) was added to the lysates containing 500 μg protein and incubated for 1.5 h at RT.

## Lentiviral particle production, neuronal infection and mCherry-D4 transfection and imaging

Packaging plasmids and short hairpin RNA plasmids against NPC1 (sh-NPC1) or a scrambled control plasmid (sh-scr; pGFP-C-shLenti) were purchased from Origene (Rockville, MD, USA). The plasmids were grown in bacteria and extracted by maxiprep (Macherey-Nagel, NucleoBond® Xtra Maxi, 740414.10). PCMV and PMD2.G (packaging plasmids) were added with polyethylenimine (PEI) to HEK295T cells, along with sh-NPC1 or sh-scr plasmids. Cell medium was replaced with Opti-MEM, and lentiviral particles were isolated by ultracentrifugation after 48 h of incubation. The pellet was resuspended in PBS and stored at −80°C until use. Lentiviral particles were added to 6 DIV neurons and incubated ON, after which the medium was replaced with fresh medium. Neurons were transfected with the mCherry-D4 plasmid [33] at 13 DIV with Lipofectamine 3000 (Life Technologies) according to the manufacturer's instructions. Z-stack images were acquired using a Nikon A1R$^+$ confocal microscope.

## Expression of GFP-wtNPC1 and D1005G NPC1 by Sindbis virus and mobility assay

The plasmid containing NPC1-EGFP was purchased from Addgene (#53521). The D1005G mutation was introduced in the Addgene plasmid by PCR. Then, wt and D1005G NPC1 were recloned into the pSinRep5 vector for Sindbis virus production and infection of organotypic cultures of hippocampal slices. After ON protein expression, slices were imaged using confocal fluorescence microscopy with an LSM800 Zeiss microscope. Images were taken every second during 30 s. The distance and velocity of the NPC1 particles were measured using ImageJ program.

## Cholesterol and 24(S)-hydroxycholesterol quantification

Prior to cholesterol quantification, total protein was measured using the Bicinchoninic Acid (BCA) Protein Assay Kit (Thermo Fisher Scientific). Cholesterol content was measured by Amplex Red Cholesterol Assay Kit (Thermo Fisher Scientific), and 24(S)-hydroxycholesterol levels were determined using the 24(S)-hydroxycholesterol enzyme-linked immunosorbent assay (ELISA) kit (Abcam, ab204530, Cambridge, UK) in accordance with the manufacturer's protocol.

## Cholesterol addition

MBCD–cholesterol complexes were purchased from Sigma-Aldrich and dissolved in medium (0.1 g MBCD-Chol in 78.63 ml medium).

Primary neuronal cultures were incubated with MBCD-Chol for 30 min. *Z*-stack images were acquired before and after the MBCD-Chol treatment using a Nikon A1R$^+$ confocal microscope.

### EFV treatment *in vitro*, *ex vivo* and *in vivo*

EFV was dissolved in DMSO at a stock concentration of 20 mM and maintained at −20°C. Synaptosomes and neuronal cultures were treated with 20 μM EFV for 2 h at 37°C. Brain slices were incubated with 20 μM EFV for 1 h prior to the field recordings, which lasted for another 2 h during which EFV was maintained. *In vivo* treatment started in mice at 6 weeks of age, with EFV administered in drinking water at a dose of 0.09 mg/kg/day as described [34]. EFV solutions were replaced in the morning of the every 3$^{rd}$ day, and control groups of mice received only vehicle (DMSO). The treatment was performed in the home cage and lasted for 8 weeks or until death for the survival experiments.

### Electrophysiological recordings in hippocampal slices

Mice were decapitated, and the brain was quickly removed and placed in ice-cold, oxygenated dissection solution (233 mM sucrose, 4 mM KCl, 5 mM $MgCl_2$, 26 mM $NaHCO_3$ and 10 mM glucose, saturated with 95% $O_2$/5% $CO_2$). Coronal slices (300 μm thick) were prepared with a vibratome (Leica, VT1200S) and kept in a chamber containing ACSF (11 mM glucose, 119 mM NaCl, 2.5 mM KCl, 1 mM $NaH_2PO_4$, 26 mM $NaHCO_3$, 1.25 mM $MgCl_2$ and 2.5 mM $CaCl_2$, saturated with 95% $O_2$/5% $CO_2$) at 32°C for at least 1 h before recording. After this recovery time, slices were maintained at 25°C. For the electrophysiological recordings, slices were perfused with ACSF at 25°C. Schaffer collateral-CA1 synaptic responses were recorded as extracellular fEPSPs from the CA1 stratum radiatum using a concentric bipolar platinum–iridium stimulation electrode and a low-resistance glass recording microelectrode filled with ACSF. pClamp9 software (Molecular Devices) was used for acquisition. To measure input–output relationship, fEPSP was evoked at different increasing stimulation intensities (from 50 up to 200 μA) and the slope of the response was calculated. This curve was also used to set the baseline fEPSP at ~20% (for paired pulse facilitation experiments), ~30% (for LTP experiments) or ~50% (for LTD experiments) of maximal slope. PPF was measured across a range of interstimulus intervals (ISIs; six traces for each ISI, 50–400 ms) at 0.067 Hz. Baseline stimulation was delivered every 15 s (0.05 ms duration pulses) for at least 20 min before LTP or LTD induction to ensure stability of the response. LTP was induced by theta burst stimulation (four pulses at 100 Hz, with the bursts repeated at 5 Hz and each tetanus including three 10-burst trains separated by 15 s). LTD was induced using 900 pulses at 1 Hz. Responses were recorded for 1 h after induction of LTP or LTD. All recordings were performed in the presence of the $GABA_A$ receptor antagonist picrotoxin (0.1 mM).

### Behavioural analysis

#### Open-field exploration test
Activity levels were recorded with a MED Associates' Activity Monitor (MED Associates, St. Albans, VT). Data were individually recorded for each animal for 5 min.

#### Object placement recognition test
Mice were subjected to three trials of a 6-min training session, during which they were allowed to freely explore 2 identical objects (small glass bottles) that were placed in defined locations of the test arena. The next day, a 6-min test session was performed, during which the position of one of the objects was changed, while the position of the other remained unaltered. A video camera was used to monitor and record the behaviour of the animals.

#### Y maze test
Mice were allowed to freely explore for 5 min in the Y maze with one of the arms closed. One hour later, the blocked arm was opened and defined as the "novel arm". Mice were placed in the Y maze in the start arm and allowed to move freely for 5 min.

#### Contextual and cued fear conditioning test
The StarFear combined system (Panlab–Harvard Apparatus) was used to test contextual and cued fear conditioning, using a protocol adapted from Ref. [53]. On the conditioning day, mice were individually placed into the conditioning chamber, and after a 3-min exploratory period, they were exposed to three tone–footshock pairings (tone-conditioned stimulus, 30 s; footshock, 2 s, 0.2 mA at the termination of the tone; separated by a 1-min intertrial interval). The following day (test day), a 6-min contextual test for conditioned fear response was conducted. On the 3$^{rd}$ day, the context and handling of the mice were changed to assess conditioned fear of the tone alone. Mice were placed in the chambers for 6 min, and immobility was assessed for a 3-min baseline period (pre-CS) followed by another 3 min for the conditioning tone (CS). Time of freezing and average motion were automatically recorded using commercial software (FREEZING V1.3, Panlab, Harvard apparatus).

#### Rotarod test
Motor testing was performed in an accelerating Rotarod apparatus (Ugo Basile, 47650 Mouse Rotarod NG), on which the mice were trained for 2 days at a constant speed: on the first day, 4 times at 4 rpm for 1 min; and on the second day, four times at 8 rpm for 1 min. On the third day, the Rotarod was set to progressively accelerate from 4 to 40 rpm for 5 min and the mice were tested four times. During the accelerating trials, the latency to fall from the rod was measured.

### Statistical analysis

The number of mice used was selected on the basis of previous phenotyping analyses conducted in the same model and calculating the statistical power of the experiment. Mice were genotyped and according to the genotype randomly assigned to the experimental groups. No outliers were excluded in the study. The information about sample collection, treatment and processing is included in Results and Material and Methods sections. Investigators assessing and measuring results were blinded to the intervention. Graphs were plotted, and statistical analyses were conducted using GraphPad Prism 5 software (GraphPad Software, USA). For comparisons between genotypes or experimental groups, unpaired two-tailed Student's *t*-tests were used for data with parametric distribution. For multiple comparisons, data with a normal distribution were analysed by two-way ANOVA followed by a Bonferroni *post*

*hoc* test. All statistical comparisons were based on biological replicates, and all values are presented as mean ± SEM. *P*-values lower than 0.05 were considered significant, and the statistical tests and sample size (*n* values) used in the experiments are specified in the Figure legends. In the figures, asterisks indicate *P*-values as follows: *$P < 0.05$; **$P < 0.01$; and ***$P < 0.001$.

**Expanded View** for this article is available online.

## Acknowledgements

We thank Profs. F.M. Platt (University of Oxford, UK) and J. Gruenberg (University of Geneva, Switzerland) for critical reading of the manuscript, the confocal and electron microscopy services of CBMSO and NB Revisions for English editing. This work was financed by grants from Fundación Alicia Koplowitz, Fundación Niemann Pick España and the Ministerio Economía, Industria y Competitividad (SAF2017-87698-R (AEI/FEDER UE)) to MDL, and by an institutional grant to the CBMSO from the Fundación Ramón Areces. DM held a fellowship from Fundación Niemann Pick España and an ERASMUS[+] grant from the University of Bonn, Germany.

## Author contributions

DNM and MDL designed the experiments and wrote the paper. DNM, GP-G, BS-H and FS performed the experiments. TK provided the mCherry-D4 DNA and advice. JAE provided equipment and advice on electrophysiological experiments and revised the manuscript.

## Conflict of interest

The authors declare that they have no conflict of interest.

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
