## [Review Process File · EMBO Reports]

NPC1 enables cholesterol mobilization during Long-Term-Potentiation that can be restored in Niemann-Pick disease type C by CYP46A1 activation

Daniel N. Mitroi, Guadalupe Pereyra-Gómez, Beatriz Soto-Huelin, Fernando Senovilla, Toshihide Kobayashi, Jose A. Esteban and María Dolores Ledesma

Review timeline:	Submission date:	21 March 2019
	Editorial Decision:	16 April 2019
	Revision received:	28 June 2019
	Editorial Decision:	6 August 2019
	Revision received:	8 August 2019
	Accepted:	23 August 2019

Editor: Martina Rembold/Esther Schnapp

Transaction Report:

1st Editorial Decision

16 April 2019

Thank you for the submission of your research manuscript to our journal. We have now received the full set of referee reports that is copied below.

As you will see, the referees acknowledge that the findings are potentially interesting. However, the referees also raise a number of - largely overlapping - concerns and have a number of suggestions on how the study could be strengthened and the conclusions be substantiated. It will be important to provide further evidence for NPC1's presence and function at the synapse and for the specificity of the antibody used. Moreover, the conclusion that NPC1 acts as a carrier for CYP46 are currently based on rather correlative data and offer limited mechanistic insight. We propose to revise the manuscript along the lines suggested by the reviewers and to substantiate all conclusions. Regarding the proposed carrier function of NPC1 for CYP46, we suggest, in agreement with the reviewers, to either provide more mechanistic insight or to remove/significantly tone down the related conclusions on NPC1's role in CYP46 transport.

To conclude, we would thus like to invite you to revise your manuscript with the understanding that the referee concerns (as detailed above and in their reports) must be fully addressed and their suggestions taken on board. Please address all referee concerns in a complete point-by-point response. Acceptance of the manuscript will depend on a positive outcome of a second round of review. It is EMBO reports policy to allow a single round of revision only and acceptance or rejection of the manuscript will therefore depend on the completeness of your responses included in the next, final version of the manuscript.

Revised manuscripts should be submitted within three months of a request for revision; they will otherwise be treated as new submissions. Please contact us if a 3-months time frame is not sufficient

for the revisions so that we can discuss the revisions further.

Supplementary/additional data: The Expanded View format, which will be displayed in the main HTML of the paper in a collapsible format, has replaced the Supplementary information. You can submit up to 5 images as Expanded View. Please follow the nomenclature Figure EV1, Figure EV2 etc. The figure legend for these should be included in the main manuscript document file in a section called Expanded View Figure Legends after the main Figure Legends section. Additional Supplementary material should be supplied as a single pdf labeled Appendix. The Appendix includes a table of content on the first page with page numbers, all figures and their legends. Please follow the nomenclature Appendix Figure Sx throughout the text and also label the figures according to this nomenclature. For more details please refer to our guide to authors.

Regarding data quantification, please ensure to specify the name of the statistical test used to generate error bars and P values, the number (n) of independent experiments underlying each data point (not replicate measures of one sample), and the test used to calculate p-values in each figure legend. Discussion of statistical methodology can be reported in the materials and methods section, but figure legends should contain a basic description of n, P and the test applied. Please also include scale bars in all microscopy images.

We now strongly encourage the publication of original source data with the aim of making primary data more accessible and transparent to the reader. The source data will be published in a separate source data file online along with the accepted manuscript and will be linked to the relevant figure. If you would like to use this opportunity, please submit the source data (for example scans of entire gels or blots, data points of graphs in an excel sheet, additional images, etc.) of your key experiments together with the revised manuscript. Please include size markers for scans of entire gels, label the scans with figure and panel number, and send one PDF file per figure.

- a complete author checklist, which you can download from our author guidelines (<http://embor.embopress.org/authorguide#revision>). Please insert page numbers in the checklist to indicate where the requested information can be found.
- a letter detailing your responses to the referee comments in Word format (.doc)
- a Microsoft Word file (.doc) of the revised manuscript text (please note the numbered reference format for EMBO reports)
- editable TIFF or EPS-formatted figure files in high resolution
(In order to avoid delays later in the publication process please check our figure guidelines before preparing the figures for your manuscript:
http://www.embopress.org/sites/default/files/EMBOPress_Figure_Guidelines_061115.pdf)
- a separate PDF file of any Supplementary information (in its final format)
- all corresponding authors are required to provide an ORCID ID for their name. Please find instructions on how to link your ORCID ID to your account in our manuscript tracking system in our Author guidelines (<http://embor.embopress.org/authorguide>).

As part of the EMBO publication's Transparent Editorial Process, EMBO reports publishes online a Review Process File to accompany accepted manuscripts. This File will be published in conjunction with your paper and will include the referee reports, your point-by-point response and all pertinent correspondence relating to the manuscript.

I look forward to seeing a revised version of your manuscript when it is ready. Please let me know if you have questions or comments regarding the revision.

REFEREE REPORTS

Referee #1:

This manuscript from the Ledesma lab reports a number of findings about Niemann-Pick C type C disease, with focus on the role of cholesterol at synapses in disease pathogenesis, the location of the NPC1 protein in neurons - also at synapses, and the results of a treatment directed at activating CYP46 (24S-cholesterol hydroxylase) as a means to mobilize cholesterol at synapses and benefit the disease.

Specific comments.

1. While there is much to consider in the various data sets of this paper, the authors focus chiefly on possible synaptic alterations in NPC1 disease, and a role for NPC1 specifically within synapses. Compromise in synaptic events of course would be anticipated in a severe pan-neuronal disease like NPC, however, as reported by the authors, few papers have focused specifically on the synapse in this condition. Importantly, the authors report finding the NPC1 protein localized to the postsynaptic density in hippocampal neurons, where it is said to be locally translated during neuronal activity (LTP induction). Unfortunately, their evidence of NPC1 in synapses using WT brain and confocal and EM rely on a single NPC1 antibody, the rabbit polyclonal ab from Novus Biologicals. Given that there is substantial published data that establish that NPC1 is a late endosomal/lysosomal transmembrane protein, it is not sufficient to use a single antibody to argue a new location for the protein. Further, the authors do not clearly explain what NPC1 would be doing at synapses, relative to what it has been shown to do at endosomes/lysosomes. Does it still interact with NPC2 at synapses, or is this a function completely independent of NPC2? If so, why do NPC1 and NPC2 diseases so closely resemble one another in both mouse models and in humans? Transmembrane proteins are notoriously difficult to generate successful antibodies to, and the NPC1 field has seen many attempts at making and using such antibodies. The current study relies solely on this one polyclonal antibody and its authenticity is not documented as part of this paper. Does it, for example, provide no staining in the NPC null model? Ruling out nuclear contamination in WBs does not rule out endosomal/lysosomal contamination. Does the Novus NPC1 ab recognize WT NPC1 and mutant NPC1 equally? Such issues could be resolved, possibly, by using the NPC1 null model in these studies.

In Fig. 8, the authors schematize NPC1 in vesicles in the synapse, not in the PSD itself, yet their single EM picture shows immunogold-NPC1 unassociated with any sort of vesicle. Some of the immunogold in this one small image also does not appear clearly to be synapse-associated. Given the great emphasis placed in this paper on a role for the NPC1 protein at synapses per se, this issue needs to be better documented (with additional antibodies and data).

2. Compromise in synapses would be anticipated in a severe pan-neuronal disease, occurring secondary to cholesterol/sphingolipid processing in the endosomal/autophagosomal and lysosomal system. Usefully, the authors do show such changes occur in hippocampal neurons in the NPC1(nmf164) model. However, the cholesterol differences, based on chemical analysis, do not appear significant in Fig. 2F. Additional details on how many samples were processed as part of this analysis would add to its believability, as would labeling with filipin or other appropriate cholesterol labeling techniques. (Indeed, later in the manuscript filipin labeling is used.) Given the abundance of unesterified cholesterol in the endosomal/lysosomal system in NPC1 mice, even minor contamination could lead to what appears to be increases in cholesterol content at the synapse. Here, clearly showing labeling of neurons at synapses, and changes in that labeling in the mutants, would overcome this significant technical shortcoming with the chemical analysis.

3. Therapy studies using the CYP46 activator, Efavirenz, are interesting. However, as with other aspects of this study, the role of this drug may not be solely through the brain-specific CYP46, as it is known to impact liver in a variety of ways (many published studies). A simple question that the authors should be able to address is what did it do to livers of the NPC1 treated mice? Did it provide benefit or further liver damage? Efavirenz is also well documented in the literature for its own

negative impact on cognition and related brain issues, in published studies of patients as well as rodents (even at lower doses). The authors need to include such frank assessments in their discussion of this drug as a treatment of NPC disease. Their concept of how it functions is also only synapse-focused. Could some other explanation be more plausible? This is worth some discussion.

4. The LTP/LTD analysis in hippocampal slices is interesting but, as with synaptic morphology changes, could be attributable to many downstream changes secondary to disease, and not necessarily resulting from a role of NPC1 at synapses.

Overall comment:

This is an interesting manuscript reporting detailed findings on synaptic structure and function in hippocampal neurons in NPC1 (nmf164) and WT mice. All of the interpretations in the manuscript are made through the lens of NPC1 being localized to synapses. Yet this has not been sufficiently and rigorously shown. Doing so would considerably elevate the importance of the findings here. Additional antibodies and mice (the null model) need to be tested, and if Fig. 8 is the correct interpretation, the identify of NPC1-labelled vesicles within synaptic spines needs to be identified, as well as shown in the immunogold studies.

Minor point.

Line 1 of the Results stats that NPC1 has been shown localized in synaptosomes but no reference is provided.

Referee #2:

The ms addresses two key questions with respect to Niemann-Pick type C disease, a rare and ultimately fatal autosomal recessive lysosomal storage disorder. Most patients present with severe neurologic and psychiatric symptoms, but it is still largely unclear why and how mutations in NPC1 or NPC2 genes damage specific neurons and how neuronal dysfunction and death can be prevented therapeutically. The authors used a well-established animal model and an impressive array of approaches ranging from biochemistry to behavioral studies. They reveal how a specific mutation in NPC1 affects synaptic transmission in the hippocampus and specific forms of plasticity. More importantly, they provide first evidence for the efficacy of a new treatment that should be rapidly testable in clinical trials alone or in combination with other approaches. Given these significant advances the manuscript is of very high interest for the community. However, there are several issues that need to be addressed. The large body of diverse approaches and results is clearly a plus, but it poses a challenge in terms of focus and calls for a more integrated presentation. More specific points are listed below:

- Title: The title appears a bit overstating and should be modified. The authors' results do not establish a causal relation between cholesterol and cognitive changes. The term "cholesterol rearrangement" is ill-defined and can be misleading (what is rearranged? the molecule itself?). It should be replaced throughout the ms by a more appropriate phrase.
- Pg. 2, para 3: The reference to Brady et al. (1966) should be replaced or complemented by Loftus et al. (1997).
- Pg. 3, para 2, Discussion: The authors should cite and probably discuss previous reports studying synaptic plasticity in NPC mouse models.
- Pg. 4, para 2. The headline should be revised. The authors' results and previous studies cited by the authors show evidence for pre- and postsynaptic localisation of NPC1. Moreover, their electron microscopic examination shows presynaptic changes and their studies of cLTP in synaptosomes cannot distinguish between pre- or postsynaptic changes. Therefore, it appears more judicious to state something like "present at synapses". The term "postsynapse" is a bit vernacular.
- Pg. 5, para 1: The time-lapse study raises the question, whether the results concern axonal or dendrite traffic. The authors should indicate which compartment was studied.
- Pg. 5, para 1: The authors should cite Gelsthorpe et al., 2008, who showed proteosomal degradation of NPC1 variants.
- Pg. 5, para 2: The authors should state whether the ultrastructural changes concern asymmetric or symmetric synapses?
- Pg. 6, para 2: The term "plasticity event" should be replaced.
- Pg. 6, para 4: The "loss of cholesterol" should be explained better: what does that mean? Where?
- Pg. 7, para 1 and elsewhere: the inappropriate use of terms "microsome" or "microsomal" should be avoided. Microsomes represent artificial fragments of the ER generated by biochemical preparation. There are no "microsomes" within cells. Cholesterol may accumulate in membrane

inclusions as shown recently. Are such inclusions present at synapses?

- Results/figure legends: the authors must state the age of the animals used for the different experiments and indicate what "n" refers to (animals, slices etc.).
- Pg. 7, para 2: The question, which carrier shuttles CYP46 to the plasma membrane, is very interesting, but it is unclear, why NPC1 should be a candidate given its "normal" location in the late endosomes/lysosomes. Moreover, one wonders whether studies of synaptosomes are really the best approach to establish presence on the "plasma membrane". Appearance of NPC1 in the plasma membrane may occur, but the authors' results do not establish this nor a carrier function for Cyp46. NPC1 may also appear on the plasma membrane after fusion of late endosomes with the plasma membrane in the course of lysosomal exocytosis. An important control experiment would be the detection of ER markers and LAMP1 at the cell surface. Clearly, this paragraph should be rewritten.
- Pg. 8, para 2: The authors should indicate for how long synaptosomes, slices etc. were treated with EFV. Would the paired-pulse facilitation be restored by longer treatment?
- Pg. 11, para 1: The authors' results do not establish that NPC1 mediates the CYP46 mobilization to the plasma membrane. Corresponding statements should be modified.
- Pg. 11, para 2: Miglustat has a number of side effects. Therefore, the corresponding statement should be modified or removed altogether.
- Pg. 11, para 2: The term "inaccessibility across" should be replaced.
- Pg. 18, para 1: In the term "at different increasing" the word "different" should be eliminated.
- Throughout the ms: With respect to the nmf164 mutation, the term "NPC1 deficiency" can be misleading. NPC1 protein is not absent, but reduced. The authors should revise their statements accordingly.
- Throughout the ms: the terms "immunofluorescence" or "immunocytofluorescence" should be replaced by more precise terms, e.g. immunocytochemical staining.
- Fig. 1A: Authors should consider to show single channel images in grey-scale for enhanced visibility.
- Fig. 1B: The size of the image could be increased. Given the notorious problems associated with antibodies against NPC1, the authors should provide (or cite) evidence showing the validity of the specific antibody that they used.
- Fig. 1C-G: Panels should be rearranged to save space.
- The few supplementary figs could be integrated in the main figures.

Referee #3:

Mitroi et al explore the role of NPC1 in the postsynaptic compartment, its role in LTP and consequences of NPC1 deficiency, as occurs in the degenerative disorder Niemann-Pick type C disease. The authors describe novel aspects of synaptic pathology in NPC mice and show that NPC1 is locally translated in synaptosomes following chemically induced LTP. Additional data suggest that NPC1 is needed for normal delivery of CYP46 and GluA1 to the plasma membrane for LTP, and suggest that pharmacological activation of CYP46 ameliorates some phenotype of NPC1 deficient mice. These findings are novel and interesting. However, a number of points need clarification to strengthen this report.

Fig 1B: Quantification of immunoEM should be provided to corroborate findings in panel 1A. Notably, some of the gold particles do not appear to be associated with membranes, an issue that should be clarified.

Fig 1C, D: If degradation is the primary determinant of NPC1 levels, why is this effect only evident in synaptosomes and not in whole brain lysates? An alternate possibility is that there is synapse loss in the NPC mouse and fewer synaptosomes are present. Is the decrease NPC1 specific or is there down regulation of many synaptic markers? Is there neuron loss at this age in NPC mice?

Fig 1G: Non-immune control is missing from ip. Also, in the left panel, why is NPC1 protein less than 180KDa, whereas in other images it runs above it?

Figs 2,3: 12 wk NPC1 mutant mice are quite impaired. Are the synaptic defects documented here developmental or degenerative? Do they progress as animals age?

Fig 2G shows that mutant synaptosomes have significantly more cholesterol than wild type, but this

observation is not supported by data in Fig 5C. Which is correct?

Fig 3I: Add number of mice to legend.

Fig 4C: The relationship between LTP and cholesterol redistribution to microsomes is unclear and authors' discussion is difficult to follow. Is this redistribution necessary for LTP? NPC1 knockdown leads to microsomal cholesterol yet mutant cells are deficient in LTP.

Fig 4D/page 7: The authors note that "an ER derived carrier interacts with the plasma membrane allowing CYP46A1 to be exposed to the outside of the cell". They further suggest "NPC1 could be the carrier mediating CYP46A1 surface expression...during LTP." How this could occur is unclear since NPC1 is located in LE/Ly, not the ER. It seems more likely that the carrier's function is influenced by NPC1-mediated cholesterol transport. Additional evidence is required to demonstrate a direct role for NPC1 in mediating CYP46A1 movement from the ER, as the authors propose.

Fig 6: Is it possible that EFV influences NPC1 trafficking, levels, or localization?

Fig 7G: Quantify diminished filipin staining and include statistical assessment of significance.

The authors note prior reports demonstrating LTD abnormalities in cerebellum of NPC1 null mice. Does EFV rescue motor impairment in mutant mice? Additionally, it would be informative to include effects on body weight and neuron loss so as to provide a clear picture of the extent of phenotypic rescue by this intervention.

The Abstract states "a common NPC1 mutation among NPC patients". However, the mouse model used contains a missense mutation not found in humans. This statement should be modified.

1st Revision - authors' response

28 June 2019

POINT-BY-POINT ANSWER TO REVIEWERS

REVIEWER 1

We thank this reviewer for considering our findings interesting

1. Importantly, the authors report finding the NPC1 protein localized to the postsynaptic density in hippocampal neurons, where it is said to be locally translated during neuronal activity (LTP induction). Unfortunately, their evidence of NPC1 in synapses using WT brain and confocal and EM rely on a single NPC1 antibody, the rabbit polyclonal ab from Novus Biologicals. Given that there is substantial published data that establish that NPC1 is a late endosomal/lysosomal transmembrane protein, it is not sufficient to use a single antibody to argue a new location for the protein.

Further, the authors do not clearly explain what NPC1 would be doing at synapses, relative to what it has been shown to do at endosomes/lysosomes. Does it still interact with NPC2 at synapses, or is this a function completely independent of NPC2? If so, why do NPC1 and NPC2 diseases so closely resemble one another in both mouse models and in humans?

Transmembrane proteins are notoriously difficult to generate successful antibodies to, and the NPC1 field has seen many attempts at making and using such antibodies. The current study relies solely on this one polyclonal antibody and its authenticity is not documented as part of this paper. Does it, for example, provide no staining in the NPC null model? Ruling out nuclear contamination in WBs does not rule out endosomal/lysosomal contamination. Does the Novus NPC1 ab recognize WT NPC1 and mutant NPC1 equally? Such issues could be resolved, possibly, by using the NPC1 null model in these studies.

In Fig. 8, the authors schematize NPC1 in vesicles in the synapse, not in the PSD itself, yet their single EM picture shows immunogold-NPC1 unassociated with any sort of vesicle. Some of the immunogold in this one small image also does not appear clearly to

be synapse-associated. Given the great emphasis placed in this paper on a role for the NPC1 protein at synapses per se, this issue needs to be better documented (with additional antibodies and data).

To address the reviewer concern about using one single antibody against NPC1 and about its specificity, we have screened a panel of antibodies and used brain extracts from NPC1 null mice. Several of the antibodies tested gave no signal in mouse samples. The only one that, besides the rabbit polyclonal from Novus Biologicals, could detect mouse NPC1 was the rabbit monoclonal from abcam #ab134113. Using this antibody, which had been validated in NPC1 null extracts, we could confirm the presence of the protein in synaptosomes and the reduction of its levels in the NPC1nmf164 mice (page 5 and new Appendix Fig S4). In addition, we confirmed the specificity of the Novus Biologicals antibody used in the original experiments by the lack of signal in extracts from NPC1 null mouse brains (page 4 and new Figure 1E).

We do not claim NPC1 function in synapses is different from that in the endolysosomal compartment. We apologize if we may have lead to misinterpretation. We propose that, thanks to its known cholesterol-mobilizing capacity, NPC1 contributes to synaptic features that were not described before. NPC1 could well be in endosomes at the postsynaptic compartment, as depicted in the model in Figure 8, and contribute to cholesterol mobilization from there. The presence of NPC1 in endosomes has been described at the presynapses by Karten et al., 2006. The fixation protocols required for the NPC1 immunoelectron microscopy analysis hinder membrane visualization, which may explain the immunogold-NPC1 unassociated with any sort of vesicle. Moreover, visualization of endosomes by electron microscopy is difficult even in fixation conditions that better keep ultrastructure due to the high dynamism of these organelles. Still, the labelling we observed is compatible with NPC1 present in membrane-like structures. This was better observed in the new electron microscope analyses using the abcam #ab134113 NPC1 antibody (page 4 and new Appendix Fig S1). We do not propose a role for NPC1 in synapses independent from NPC2. The two proteins could jointly act to mobilize cholesterol as it has been described in the endolysosomal compartment. Supporting this view, and encouraged by this reviewer comment, we have detected NPC2 at the postsynapses by immunoelectron microscopy and in synaptosomes by Western blot (page 12 and new Appendix Fig S12). Different from NPC1, the synaptic levels of NPC2 are not altered in the NPC1nmf164 compared to wt mice.

2. Compromise in synapses would be anticipated in a severe pan-neuronal disease, occurring secondary to cholesterol/sphingolipid processing in the endosomal/autophagosomal and lysosomal system. Usefully, the authors do show such changes occur in hippocampal neurons in the NPC1(nmf164) model. However, the cholesterol differences, based on chemical analysis, do not appear significant in Fig. 2G. Additional details on how many samples were processed as part of this analysis would add to its believability, as would labeling with filipin or other appropriate cholesterol labeling techniques. (Indeed, later in the manuscript filipin labeling is used.) Given the abundance of unesterified cholesterol in the endosomal/lysosomal system in NPC1 mice, even minor contamination could lead to what appears to be increases in cholesterol content at the synapse. Here, clearly showing labeling of neurons at synapses, and changes in that labeling in the mutants, would overcome this significant technical shortcoming with the chemical analysis.

To satisfy this reviewer query we performed co-staining experiments of Filipin and the synaptic marker PSD95 in the hippocampus of wt and NPC1nmf164 mice (Figure A for this reviewer only). Unfortunately, and although the increased Filipin staining is evident in the NPC1nmf164 mice reflecting the higher cholesterol content, the different Filipin and PSD95 patterns limited accurate quantification and prevented any solid conclusion on synaptic cholesterol levels from this colocalization study. Fluorescent probes available to detect cholesterol such as Filipin have limited capacity to detect subtle cholesterol changes. We believe chemical analysis of cholesterol by enzymatic assays is a more accurate and reliable method. The Amplex Red Cholesterol assay kit we have used in synaptosomes is a widely

used method that, relying on cholesterol oxidation, can detect low concentrations of this lipid. This method indicated a mild, yet statistical significant, 16.5% increase of cholesterol in synaptosomes from *NPC1nmf164* compared to wt mice (page 6). The data shown in Figure 2G correspond to synaptosomes isolated from 4 wt and 4 *NPC1nmf164* mice and the Student-t-test indicated a significant P value=0.0435. This is now clarified in the Figure legend.

Figure A for this reviewer only. Representative images of the fluorescent cholesterol-binding probe Filipin and of immunocytochemical staining against the synaptic marker PSD95 in CA1 hippocampal tissue from wt and *NPC1nmf164* mice of 3 months of age. DAPI stains cell nuclei

3. Therapy studies using the CYP46 activator, Efavirenz, are interesting. However, as with other aspects of this study, the role of this drug may not be solely through the brain-specific CYP46, as it is known to impact liver in a variety of ways (many published studies). A simple question that the authors should be able to address is what did it do to livers of the NPC1 treated mice? Did it provide benefit or further liver damage? Efavirenz is also well documented in the literature for its own negative impact on cognition and related brain issues, in published studies of patients as well as rodents (even at lower doses). The authors need to include such frank assessments in their discussion of this drug as a treatment of NPC disease. Their concept of how it functions is also only synapse-focused. Could some other explanation be more plausible? This is worth some discussion.

The reviewer is right that, although rare, cases of hepatotoxicity have been described after chronic Efavirenz treatment. However, our observation that Efavirenz did not have evident effects in wt mice and improved weight gain and expanded life span in the *NPC1nmf164* mice would argue against further liver damage after this treatment. Following this reviewer suggestion we have analysed the impact of Efavirenz treatment in the liver. We measured inflammation by immunostaining against the macrophage marker F4/80 and cholesterol levels by Filipin and by Amplex Red. We did not find differences in the number and area of macrophages (which is increased in the non-treated *NPC1nmf164* mice supporting inflammation) or in the cholesterol levels (pages 10 and 13 and new Appendix Fig S11). This supports that the Efavirenz effects on neuroinflammation and cholesterol observed in brain are mediated by CYP46A1, which is a brain specific enzyme not present in the liver. On the other hand, neurological side effects, such as depression and cognitive deficits, have been described in certain HIV patients chronically treated with Efavirenz. We believe our data help to understand these effects. Efavirenz would enhance CYP46A1 activity reducing cholesterol levels in brains that, different from the NPC patients, do not have an excess of this lipid. Since cholesterol is necessary for LTP its abnormal reduction could impair synaptic plasticity contributing to the neurological alterations described in certain HIV patients. This is now discussed in pages 12-13.

4. The LTP/LTD analysis in hippocampal slices is interesting but, as with synaptic morphology changes, could be attributable to many downstream changes secondary to disease, and not necessarily resulting from a role of NPC1 at synapses.

We agree with the reviewer that other downstream changes cannot be ruled out. However, we believe that the discovery of NPC1 mRNA and of its LTP-induced expression at synapses (Figures 1F, G), together with the impaired AMPARc surface delivery in NPC1 silenced cultured neurons (Figure 4F) argue in favour of NPC1 influencing specific synaptic features.

Minor point.

Line 1 of the Results states that NPC1 has been shown localized in synaptosomes but no reference is provided.

Karten and coauthors (J Lipid Res 2006) showed the presence of NPC1 in synaptosomes derived from mouse cerebellum. The reference is provided in the introduction (page 3) and in the first line of Results (page 4).

REVIEWER 2

We thank this reviewer for acknowledging the variety of approaches ranging from biochemistry to behavioural studies we have used in our study and for considering our results a significant advance of very high interest for the community.

Title: The title appears a bit overstating and should be modified. The authors' results do not establish a causal relation between cholesterol and cognitive changes. The term "cholesterol rearrangement" is ill-defined and can be misleading (what is rearranged? the molecule itself?). It should be replaced throughout the ms by a more appropriate phrase.

Following this reviewer suggestion we have changed the title to:

"NPC1 enables cholesterol mobilization during long-term potentiation that can be restored in Niemann-Pick disease type C by CYP46A1 activation".

We have also replaced throughout the manuscript the term "rearrangement" for "mobilization" or "redistribution". We agree with the reviewer this better reflects the cholesterol changes observed and thank him/her for the suggestion

- Pg. 2, para 3: The reference to Brady et al. (1966) should be replaced or complemented by Loftus et al. (1997).

Loftus et al (1997) is now cited in page 2.

- Pg. 3, para 2, Discussion: The authors should cite and probably discuss previous reports studying synaptic plasticity in NPC mouse models.

As already mentioned in the introduction Sun and collaborators (Cerebellum, 2011) described impaired LTD in the cerebellum of NPC1 null mice and linked it to decreased ATP/adenosine release. Zhou and collaborators (Hippocampus, 2010) also attributed to reduced extracellular adenosine levels the deficient LTP they described in hippocampal slices of NPC1 null mice. During the revision process of this manuscript another study (Vivas et al., Cell Reports May 2019) has used NPC1 mutant mice to describe a link between lysosomal cholesterol and PtdIns(4,5)P2 that influences neuronal excitability. These reports are now discussed in page 11.

- Pg. 4, para 2. The headline should be revised. The authors' results and previous studies cited by the authors show evidence for pre- and postsynaptic localisation of NPC1. Moreover, their electron microscopic examination shows presynaptic changes and their studies of cLTP in synaptosomes cannot distinguish between pre- or postsynaptic changes. Therefore, it appears more judicious to state something like "present at synapses". The term "postsynapse" is a bit vernacular.

Following this reviewer suggestion we now state "present at the synapses" in the

headline in page 4

- Pg. 5, para 1: *The time-lapse study raises the question, whether the results concern axonal or dendrite traffic. The authors should indicate which compartment was studied. Unfortunately, the time-lapse studies do not allow an accurate distinction between axons and dendrites. We can only state the compartments studied in these experiments are the neurites and therefore refer now to neurite traffic (page 5)*

- Pg. 5, para 1: *The authors should cite Gelsthorpe et al., 2008, who showed proteosomal degradation of NPC1 variants. Gelsthorpe et al (2008) is now cited in page 5.*

- Pg. 5, para 2: *The authors should state whether the ultrastructural changes concern asymmetric or symmetric synapses? We have focused on asymmetric excitatory synapses of the hippocampal CA1 characterized by thick PSD and the presence of round-shaped vesicles in presynaptic terminals. This is now clarified in page 5.*

- Pg. 6, para 2: *The term "plasticity event" should be replaced. This term has been replaced by "form of plasticity"*

- Pg. 6, para 4: *The "loss of cholesterol" should be explained better: what does that mean? Where? Loss of cholesterol means a reduction of cholesterol amount in the 100000g pellet from slices. This pellet is enriched in plasma membrane. This is now better explained in page 7.*

- Pg. 7, para 1 and elsewhere: *the inappropriate use of terms "microsome" or "microsomal" should be avoided. Microsomes represent artificial fragments of the ER generated by biochemical preparation. There are no "microsomes" within cells. Cholesterol may accumulate in membrane inclusions as shown recently. Are such inclusions present at synapses? We apologize for having used, following the terms used in Brachet et al., J Cell Biol 2015, the word microsomes inappropriately. By this term we referred to light membranes that remain in the supernatant after a 100000g centrifugation. This is now clarified in page 7 and the term microsome has been changed to vesicular or light membrane fractions.*

- Results/figure legends: *the authors must state the age of the animals used for the different experiments and indicate what "n" refers to (animals, slices etc.). Age of the animals and what n refers to are now indicated in each Figure legend*

- Pg. 7, para 2: *The question, which carrier shuttles CYP46 to the plasma membrane, is very interesting, but it is unclear, why NPC1 should be a candidate given its "normal" location in the late endosomes/lysosomes. Moreover, one wonders whether studies of synaptosomes are really the best approach to establish presence on the "plasma membrane". Appearance of NPC1 in the plasma membrane may occur, but the authors' results do not establish this nor a carrier function for Cyp46. NPC1 may also appear on the plasma membrane after fusion of late endosomes with the plasma membrane in the course of lysosomal exocytosis. An important control experiment would be the detection of ER markers and LAMP1 at the cell surface. Clearly, this paragraph should be rewritten.*

While we agree with this reviewer that synaptosomal preparations have technical limitations we used them since they allow the specific analysis of synaptic membranes. Biotinylation assays in synaptosomes, as the ones performed for CYP46A1 in Figure 4E, have been used in previous studies to demonstrate the presence of proteins at the synaptic plasma membrane (i.e. Olivero et al., Mol Neurobiol 2019; Qiao et al., Neuroscience 2017; Okuda et al., J neurosci 2011). Following this reviewer suggestion we have conducted biotinylation assays in synaptosomes for ER and endolysosomal markers such as Calnexin and Lamp1,

respectively (Figure B for this reviewer only). We detect Calnexin at the synaptic surface in both wt and *NPC1^{nmf164}* mice supporting the possibility that the ER protein CYP46A1 reaches the synaptic surface by ER-plasma membrane contact. We also detect the late endosome/lysosome marker Lamp1 at the surface, particularly in *NPC1^{nmf164}* mice where lysosomes are more abundant. This would be compatible with NPC1 reaching the plasma membrane in the course of endolysosomal exocytosis as the reviewer suggests. However, the results obtained do not allow us to conclude that Calnexin and lysosomal markers are more abundant in the synaptic plasma membrane of wt vs *NPC1^{nmf164}* mice. Further analysis will be necessary to determine the exact mechanism by which NPC1 facilitates CYP46A1 shuttling to the plasma membrane. It is possible that NPC1 reaches the plasma membrane from endolysosomal compartments and there act as a tether with the ER, thus facilitating the surface delivery of the ER protein CYP46A1. The paragraph has been rewritten accordingly in page 11 and in the Figure 8 legend.

Figure B for this reviewer only. Western blots against LAMP1, Calnexin (CANX) and actin- β (ACTB) in total extracts (T) and biotin-streptavidin immunoprecipitates (S) from synaptosomes from wt and *NPC1^{nmf164}* mice in which cLTP was induced or not (n=2 mice, 3 month-old)

- Pg. 8, para 2: The authors should indicate for how long synaptosomes, slices etc. were treated with EFV. Would the paired-pulse facilitation be restored by longer treatment?

Time of incubation of the different preparations with EFV is indicated in the Materials and Methods section under the subtitle “EFV treatment *in vitro*, *ex vivo* and *in vivo*” in the main text and also in the figure legends. To answer this reviewer query about the paired-pulse facilitation we extended the incubation of slices with EFV to 4 hours (New Appendix Fig S8) observing no changes in the effects compared to the 1hour-long incubation.

- Pg. 11, para 1: The authors' results do not establish that NPC1 mediates the CYP46 mobilization to the plasma membrane. Corresponding statements should be modified. Our results show that when NPC1 is mutated the presence of CYP46A1 at the plasma membrane upon LTP induction is reduced. As mentioned above, we agree with the reviewer that further work is necessary to establish the exact mechanism by which NPC1 influences CYP46A1 mobilization to the plasma membrane. We have modified the corresponding statements (page 11)

- Pg. 11, para 2: Miglustat has a number of side effects. Therefore, the corresponding

statement should be modified or removed altogether.

The statement has been modified (page 12)

- Pg. 11, para 2: The term "inaccessibility across" should be replaced.

"Inaccessibility across the blood-brain barrier" has been replaced by "poor penetration through the blood brain barrier" (page 13).

- Pg. 18, para 1: In the term "at different increasing" the word "different" should be eliminated.

The word "different" has been eliminated.

*- Throughout the ms: With respect to the *nmf164* mutation, the term "NPC1 deficiency" can be misleading. NPC1 protein is not absent, but reduced. The authors should revise their statements accordingly.*

We used deficiency meaning shortage, scarcity. The statements have been revised.

- Throughout the ms: the terms "immunofluorescence" or "immunocytofluorescence" should be replaced by more precise terms, e.g. immunocytochemical staining.

We have used immunocytochemical staining throughout the manuscript

- Fig. 1A: Authors should consider to show single channel images in grey-scale for enhanced visibility.

We now show single channel images in grey-scale

- Fig. 1B: The size of the image could be increased. Given the notorious problems associated with antibodies against NPC1, the authors should provide (or cite) evidence showing the validity of the specific antibody that they used.

Following this reviewer suggestion the size of the electron microscopy image in Figure 1B has been increased and the pre and postsynaptic compartments highlighted by different colours.

To address the reviewer concern about the validity and specificity of the antibody against NPC1, we have screened a panel of antibodies and used brain extracts from *NPC1 null* mice. Several of the antibodies tested gave no signal in mouse samples. The only one that, besides the rabbit polyclonal from Novus Biologicals, could detect mouse NPC1 was the rabbit monoclonal from abcam #ab134113, which had been validated in *NPC1 null* extracts. Using this antibody we could confirm the presence of the protein in synaptosomes and the reduction of its levels in *NPC1nmf164* mice (page 5 and new Appendix Fig S4). In addition, the lack of signal in extracts from *NPC1 null* mouse brains confirmed the specificity of the Novus Biologicals antibody used in the original experiments (New Figure 1E). We have also performed new immunoelectron microscopy analysis with the abcam #ab134113 antibody. Images with higher magnification showing NPC1 in membrane-like structures at the postsynapse are provided in the new Appendix Fig S1.

- Fig. 1C-G: Panels should be rearranged to save space.

The panels have been rearranged and a new panel 1E has been added with new data.

- The few supplementary figs could be integrated in the main figures.

In this revised version we have 12 Supplementary Figures, which we kindly ask to keep for the sake of clarity and not to overload the main figures.

REVIEWER 3

We thank the reviewer for considering our findings novel and interesting.

Fig 1B: Quantification of immunoEM should be provided to corroborate findings in panel 1A. Notably, some of the gold particles do not appear to be associated with

membranes, an issue that should be clarified.

The fixation protocols required for NPC1 immunoelectron microscopy analysis hinder membrane visualization, which may explain the immunogold-NPC1 unassociated with any sort of vesicle. The highly dynamic nature of endosomes compared to synaptic vesicles renders difficult their visualization by electron microscopy. Still, the labelling we observed is compatible with NPC1 present in membrane-like structures. This was better observed in the new electron microscope analyses using the abcam #ab134113 NPC1 antibody (page 4 and new Appendix Fig S1). Quantification of immunoelectron micrographs is not very reliable since the sensitivity of this technique is low. This is the reason why we have used immunocytochemistry (Fig 1A, B) and Western blot of synaptosomes (Fig 1D and new Appendix Fig S4) for quantifications and hope the reviewer accepts these quantitative means.

Fig 1C, D: If degradation is the primary determinant of NPC1 levels, why is this effect only evident in synaptosomes and not in whole brain lysates? An alternate possibility is that there is synapse loss in the NPC mouse and fewer synaptosomes are present. Is the decrease NPC1 specific or is there down regulation of many synaptic markers? Is there neuron loss at this age in NPC mice?

It is important to take into account that whole brain lysates would show the contribution not only of neurons but also glial cells where NPC1 is also expressed and may compensate for reduced levels in neurons. It could also be that the effect is more evident in synaptosomes because NPC1, being locally translated, could undergo enhanced degradation at this specific cellular site. Synapse loss indeed occurs in the *NPC1nmf164* mice (see Figure 2B). However, this effect should not influence the results in our experimental setting since Western blot analysis in synaptosomes was always normalized by the total amount of protein (confirmed by ACTB or tubulin loading controls). Moreover, we found no down regulation of other synaptic proteins like PSD95, Synaptophysin, Syntaxin 1A or VAMP2, arguing in favour of a specific feature of NPC1 (page 5 and new Appendix Fig S5). To address this reviewer query on neuronal loss we analysed Purkinje cells in the cerebellum by Calbindin staining and neurons in the cortex and hippocampus by NeuN staining. At the age analysed we found Purkinje cell loss but no significant death in the cortex and hippocampus of *NPC1nmf164* mice compared to wt (page 9 and New Appendix Fig S10).

Fig 1G: Non-immune control is missing from ip. Also, in the left panel, why is NPC1 protein less than 180KDa, whereas in other images it runs above it?

We now show the non-immune control for the IP in Figure 1H. We apologize for the mistake in the molecular weight marker that has now been corrected in this Figure.

Figs 2,3: 12 wk NPC1 mutant mice are quite impaired. Are the synaptic defects documented here developmental or degenerative? Do they progress as animals age? As indicated in Maue et al Hum Mol Gen, 2012 (also confirmed by our own observations) NPC1nmf164 mice are born apparently normal. With age they show a progressive degenerative course, which is slower than in NPC1 null mice.

Fig 2G shows that mutant synaptosomes have significantly more cholesterol than wild type, but this observation is not supported by data in Fig 5C. Which is correct? The differences between these figures can be explained by the different controls used as reference values. In Fig 2G the amount of cholesterol in the NPC1nmf164 synaptosomes is referred to that in wt mice, which is considered 100%. In contrast, cholesterol amount in the untreated samples for each condition (wt or NPC) are considered 100% in Figure 5C. This is now clarified in the Figure legends.

Fig 3I: Add number of mice to legend.
Number of mice has been added

Fig 4C: The relationship between LTP and cholesterol redistribution to microsomes is

unclear and authors' discussion is difficult to follow. Is this redistribution necessary for LTP? NPC1 knockdown leads to microsomal cholesterol yet mutant cells are deficient in LTP.

According to Brachet et al. J Cell Biol 2015, LTP induces cholesterol redistribution and reduction that are necessary for its progression. Using mCherry-D4 expression to label cholesterol they showed a change from diffuse to clustered distribution upon LTP in hippocampal slices from wt mice. In Figure 4C we show that in neurons where NPC1 expression has been down regulated this shift is not achieved since mCherry-D4 cholesterol staining is already clustered at basal levels.

Fig 4D/page 7: The authors note that "an ER derived carrier interacts with the plasma membrane allowing CYP46A1 to be exposed to the outside of the cell". They further suggest "NPC1 could be the carrier mediating CYP46A1 surface expression...during LTP." How this could occur is unclear since NPC1 is located in LE/Ly, not the ER. It seems more likely that the carrier's function is influenced by NPC1-mediated cholesterol transport. Additional evidence is required to demonstrate a direct role for NPC1 in mediating CYP46A1 movement from the ER, as the authors propose.

We agree with this reviewer that we cannot conclude on the exact mechanism by which NPC1 facilitates CYP46 exposure to the plasma membrane. It is indeed possible that another carrier is influenced by NPC1-mediated cholesterol transport. It could also be that NPC1 reaches the plasma membrane from the endolysosomal compartments and there acts as a tether facilitating ER-plasma membrane contact thus enabling the surface delivery of CYP46A1. This is now discussed in page 11 and in Figure 8 legend.

Fig 6: Is it possible that EFV influences NPC1 trafficking, levels, or localization?

To address this question we have performed immunofluorescence against NPC1 in primary cultured neurons from wt mice treated or not with EFV. We do not observe changes on the levels or distribution of the protein after treatment (page 9 and new Appendix Fig S9).

Fig 7G: Quantify diminished filipin staining and include statistical assessment of significance.

Quantification has been performed and the data with the statistical analysis are included in the new Figure 7H.

The authors note prior reports demonstrating LTD abnormalities in cerebellum of NPC1 null mice. Does EFV rescue motor impairment in mutant mice? Additionally, it would be informative to include effects on body weight and neuron loss so as to provide a clear picture of the extent of phenotypic rescue by this intervention.

We now provide with data showing the improvement of motor abilities (assessed by the Rotarod test) in EFV treated NPC1^{Inmf164} mice (page 9 and New Appendix Fig S10). This could be explained by the prevention of Purkinje cell loss in the cerebellum (New Appendix Fig S10). Data showing improvement in body weight upon EFV treatment are also included in the new Figure 7B.

The Abstract states "a common NPC1 mutation among NPC patients". However, the mouse model used contains a missense mutation not found in humans. This statement should be modified.

The statement has been changed to "a mutation in a region of the NPC1 gene commonly altered in NPC patients".

Thank you for the submission of your revised manuscript. I have taken over its handling to save some time, as Martina is currently not in the office. We have now received the enclosed comments from the referees, and I am happy to tell you that we can in principle accept your manuscript now. Only a few more minor changes from the editorial side are still required

REFEREE REPORTS

Referee #2:

The authors have responded to this referee's comments in a satisfactory manner and the revised version appears now suitable for publication in EMBO reports.

Referee #3:

The authors have adequately addressed the points raised in the prior review.

2nd Revision - authors' response

8 August 2019

The authors performed all minor editorial changes.

Corresponding Author Name: MARIA DOLORES LEDESMA
Manuscript Number: EMBOR-2019-48143